# Fine-Tuning Masked Diffusion for Provable Self-Correction

Jaeyeon Kim [* 1]   Seunggeun Kim [* 2]   Taekyun Lee [* 2]   David Z. Pan [2]   Hyeji Kim [2]   Sham Kakade [1 3]   Sitan Chen [1]

## Abstract

A natural desideratum for generative models is *self-correction*–detecting and revising low-quality tokens at inference. While Masked Diffusion Models (MDMs) have emerged as a promising approach for generative modeling in discrete spaces, their capacity for self-correction remains poorly understood. Prior attempts to incorporate self-correction into MDMs either require overhauling MDM architectures/training or rely on imprecise proxies for token quality, limiting their applicability. Motivated by this, we introduce PRISM–**P**lug-in **R**emasking for **I**nference-time **S**elf-correction of **M**asked Diffusions–a lightweight, model-agnostic approach that applies to any pretrained MDM. Theoretically, PRISM defines a self-correction loss that provably learns per-token quality scores, without RL or a verifier. These quality scores are computed in the same forward pass with MDM and used to detect low-quality tokens. Empirically, PRISM advances MDM inference across domains and scales: Sudoku; unconditional text (170M); and code with LLaDA (8B). We open-source our codebase in (Link).

## 1. Introduction

Masked Diffusion Models (MDMs) (Sahoo et al., 2024; Gat et al., 2024; Shi et al., 2024) have emerged as a promising approach for generative modeling in discrete domains. At inference time, they start from a fully masked sequence and gradually unmask the masked tokens in arbitrary order to obtain a clean sequence. This flexibility in generation order, combined with recent approaches to deployment at scale (Nie et al., 2025; Ye et al., 2025; DeepMind, 2025; Labs et al., 2025; Gong et al., 2025), has enabled them to surpass their autoregressive counterparts on several downstream tasks, such as reasoning, coding, and planning.

However, the standard MDM design lacks the ability to self-correct–*a natural desideratum for generative models*—that is, identifying low-quality or incorrect tokens and correcting them at inference. This limitation is particularly crucial in MDMs due to their parallel, few step inference nature. Parallelized inference with only a few steps makes MDMs more susceptible to dependency errors across positions (Liu et al., 2024a; Xu et al., 2024; Kang et al., 2025), as they model per-position unmasking posteriors rather than the full joint distribution over positions.

Several recent works have attempted to overcome this limitation in MDMs (Campbell et al., 2022; Lezama et al., 2022b; Zhao et al., 2024; Wang et al., 2025; von Rütte et al., 2025; Peng et al., 2025). However, these approaches come with one of two drawbacks: they either **(1)** indirectly or inaccurately learn a notion of token quality and are potentially inefficient to evaluate or **(2)** entirely change the MDM framework or architecture, limiting their applicability.

**Contribution.** In this work, we propose a simple, principled solution addressing these limitations, **PRISM**: **P**lug-in **R**emasking for **I**nference-time **S**elf-correction of **M**asked diffusions, a plug-and-play fine-tuning framework that easily adapts to any pretrained MDM (addressing **(1)**) and directly learns the clean token's quality given a partially masked sequence (addressing **(2)**).

**Theoretically**, PRISM introduces a novel self-correction loss that *provably* learns the per-token quality scores. It attaches a lightweight adapter to any pretrained MDM and fine-tunes with this loss; at inference, the learned scores detect low-quality tokens to *remask* and potentially be revised in subsequent inference steps (Figure 1, right). **Practically**, PRISM improves MDM inference across a range of domains and scales: Sudoku, unconditional text generation with a 170M MDM, and code generation with LLaDA (Nie et al., 2025), an open-source 8B MDM. Notably, on LLaDA, with $< 500$ GPU-hours of fine-tuning, it acquires the ability to self-correct, outperforming baselines on MBPP (Austin et al., 2021b) and HumanEval (Chen, 2021).

**Organization.** We begin by reviewing MDMs in Section 2.1 and previous work on equipping MDMs with self-correction

---

[*] Jaeyeon Kim and Seunggeun Kim contributed equally; Taekyun Lee is also a co–first author. [1]Harvard University [2]The University of Texas at Austin [3]Kempner Institute. Correspondence to: Jaeyeon Kim <jaeyeon_kim@g.harvard.edu>.

*Proceedings of the 43rd International Conference on Machine Learning*, Seoul, South Korea. PMLR 306, 2026. Copyright 2026 by the author(s).

*Figure 1.* **PRISM**. (**Left**) MDM learns *unmasking posterior* to unmask tokens, after which they remain fixed. (**Middle**) PRISM fine-tuning introduces *per-token quality*, which is used to detect incorrect tokens and *remask* them. At inference, the fine-tuned MDM jointly computes *unmasking posterior* and *per-token quality*, respectively performing unmasking and remasking. (**Right**) PRISM training pair generation: sample a masked sequence **z** from **x**; from **z**, unmask a chosen position using the pretrained MDM to obtain **y**.

ability in Section 2.2. We introduce PRISM in Section 3 with its theoretical foundation and validate its empirical ability in Section 4.

**Concurrent work.** Meshchaninov et al. (2025); Li et al. (2025); Schiff et al. (2026), while differing in technical details, also address the self-correction problem for MDMs by either training an additional module or fine-tuning the MDM backbone that takes $\hat{\mathbf{x}}$ as input, thus falling under the class of proxy-$\hat{\mathbf{x}}$ approaches described in Section 2.2. Most closely related, (Huang et al., 2025) shares the motivation and experimental setup. More recently, Bie et al. (2026) trains large-scale MDMs at 16B and 100B parameters. Although the details are not disclosed, their codebase suggests that they learn a per-position distribution rather than a scalar. We return to this discussion in Section 5.[1]

## 2. Preliminaries

### 2.1. Masked Diffusion Models

**Notation.** Assume we aim to learn the distribution over sequences of length $L$ with vocabulary $\mathcal{V}$, namely the true distribution $\mathbf{x} \sim p_{\text{data}}$. $\mathbf{x}^i$ denotes the $i$-th element of a given sequence $\mathbf{x} = (\mathbf{x}^1, \ldots, \mathbf{x}^L)$ and $\Delta(\mathcal{V})$ indicates the simplex of probability distributions over $\mathcal{V}$.

**MDM training.** Although MDMs can be interpreted from several viewpoints, we adopt an *any-order* language model viewpoint (Ou et al., 2024; Zheng et al., 2024), which is essential to understanding our PRISM framework. Roughly speaking, MDM learns the posterior marginals over clean tokens in each masked position, conditioned on a partially masked sequence. We now formalize how this objective is

---

[1] This paragraph has been included since the second version of this manuscript.

defined and trained. To *learn* the posterior, MDM performs a *masking process* using an auxiliary mask token $\mathbf{m}$. For a given $\mathbf{x} \sim p_{\text{data}}$, this masking process samples a partially masked sequence $\mathbf{z} \in (\mathcal{V} \cup \{\mathbf{m}\})^L$ defined in two equivalent ways (see Appendix B.2 for equivalence):

(1) Draw $n \sim \text{Unif}\{0, \ldots, L\}$ and replace the tokens at uniformly selected $n$ indices in $\mathbf{x}$ with $\mathbf{m}$.

(2) Draw $t \sim \text{Unif}[0, 1]$ and for each $i$, replace $\mathbf{x}^i \in \mathcal{V}$ with $\mathbf{m}$ with probability $t$.

To clarify, in (1), for a given $n$, we uniformly sample an index set from all size-$n$ subsets of $\{1, \ldots, L\}$, and mask corresponding positions of $\mathbf{x}$. This masking process defines a random variable $\mathbf{z}$ conditioned on $\mathbf{x}$, and marginalizing it over $\mathbf{x} \sim p_{\text{data}}$ induces a joint distribution over $(\mathbf{x}, \mathbf{z})$.

We refer to the resulting coordinate-wise posterior conditioned on $\mathbf{z}$ as the *unmasking posterior*; $p(\mathbf{x}^i = \cdot \mid \mathbf{z})$, i.e., the distribution over clean tokens at position $i$ given a partially masked sequence $\mathbf{z}$. This unmasking posterior is the core modeling component in MDM and is parameterized by a neural network $f_\theta$, which takes $\mathbf{z}$ as an input and outputs a tensor of shape $|\mathcal{V}| \times L$. Its $i$-th column, $f_\theta^i(\cdot \mid \mathbf{z}) \in \Delta(\mathcal{V})$, models the unmasking posterior $f_\theta^i(v \mid \mathbf{z}) \approx p(\mathbf{x}^i = v \mid \mathbf{z})$. To train $f_\theta$, MDM minimizes the following loss:

$$\mathcal{L}(\theta) := \mathbb{E}_{\mathbf{x}, \mathbf{z}} \left[ \frac{1}{n} \sum_{i: \, \mathbf{z}^i = \mathbf{m}} -\log f_\theta^i(\mathbf{x}^i \mid \mathbf{z}) \right], \quad (1)$$

which is the cross-entropy loss summed over all masked indices. Here, the expectation is taken over the joint distribution of $(\mathbf{x}, \mathbf{z})$ defined above. As desired, the unique minimizer $f_{\theta^\star}$ of the loss $\mathcal{L}_\theta$ satisfies $f_{\theta^\star}^i(v \mid \mathbf{z}) = p(\mathbf{x}^i = v \mid \mathbf{z})$. This connects the MDM training loss to any-order language models such as BERT (Devlin et al., 2019), which model

the posterior marginals at all masked positions. We note that the training loss (1) allows other equivalent formulations, as developed in previous MDM works (Nie et al., 2025; Sahoo et al., 2024; Shi et al., 2024).

**MDM inference.** MDM inference starts from a length-$L$ masked sequence $\mathbf{x}_1 = (\mathbf{m}, \dots, \mathbf{m})$ and proceeds over a monotonically decreasing sequence of times $t_0 = 1 > \cdots > t_N = 0$. At each step $t_\ell$, given partially masked sequence $\mathbf{x}_{t_\ell} \in (\mathcal{V} \cup \{\mathbf{m}\})^L$, we proceed by two steps to obtain $\mathbf{x}_{t_{\ell+1}}$:

(a) Choose a subset of masked tokens $\mathcal{S}$ to unmask, $\mathcal{S} \subseteq \{i \mid \mathbf{x}_{t_\ell}^i = \mathbf{m}\}$.

(b) For each $i \in \mathcal{S}$, unmask $\mathbf{x}_{t_\ell}^i$ to a clean token sampled from $f_\theta^i(\cdot \mid \mathbf{x}_{t_\ell}) \in \Delta(\mathcal{V})$.

Notably, step (a)—the choice of $\mathcal{S}$—is highly flexible and central to MDM's gains on downstream tasks; $\mathcal{S}$ may be chosen arbitrarily. Two predominant strategies are independently including each masked index $\mathbf{x}_{t_\ell}^i = \mathbf{m}$ with a fixed probability (Sahoo et al., 2024; Shi et al., 2024) or adaptively selecting indices, e.g., the most confident tokens or the leftmost tokens (Chang et al., 2022; Zheng et al., 2023; Kim et al., 2025; Nie et al., 2025).

### 2.2. Self-correction for MDM via Per-token Quality

An ideal generative model would detect low-quality tokens at inference time and correct them when appropriate. Prior work has explored self-correction in LLMs, including (Kumar et al., 2024; Gou et al., 2023), and, recently, in MDMs. Despite varying instantiations, these methods for MDM can be organized into a two-step recipe. At each inference step, given $\mathbf{x}_{t_\ell}$:

(a) Compute a per-token "quality score" for each $\mathbf{x}_{t_\ell}^i \neq \mathbf{m}$ and, according to it, select a subset of clean tokens $\mathcal{T}$ to remask.

(b) For each $i \in \mathcal{T}$, remask $\mathbf{x}_{t_\ell}^i$ and either replace it with a new clean or leave it masked.

At step (a), the set $\mathcal{T}$ is typically chosen with the lowest per-token quality, optionally with stochasticity. At step (b), the remasked $\mathbf{x}_{t_\ell}^i$ can be immediately resampled (using the pretrained model or an auxiliary module) or kept to engage further for future unmasking/remasking steps. Combined with the standard MDM unmasking step (Section 2.1), inference alternates between unmasking and (re)masking; these may be interleaved or staged, depending on the method.

**Challenge.** The key challenges are twofold: formulating a **principled definition of per-token quality** and **constructing a model** that estimates it. These can be formalized as follows: A per-token quality is a function $g_\star$

that takes any partially masked sequence $\mathbf{y}$ and returns position-wise scalars for non-masked positions $\mathbf{y}^i \neq \mathbf{m}$: $g_\star : (\mathcal{V} \cup \{\mathbf{m}\})^L \to [0,1]^L$,

$$g_\star^i(\mathbf{y}) \approx (\text{Per-token quality}) = (how\ likely\ \mathbf{y}^i\ \text{is given}\ \mathbf{y}).$$

The phrase "how likely" is a modeling choice; whatever the choice, it should align with the intended notion of per-token quality (our PRISM instantiation is given in Section 3). The goal is a computationally efficient algorithm for modeling $g_\star$ that, in the limit, provably learns the chosen definition of per-token quality. As we argue next, prior work typically lacks either a precise, testable target of per-token quality or entirely changes the MDM design itself, limiting its applicability. We therefore organize the prior work along whether it specifies a precise target for per-token quality on the state during inference $\mathbf{x}_{t_j}$ and how disruptive it is to the MDM design.

- Training-free approaches: DFM (Gat et al., 2024), ReMDM (Wang et al., 2025), P2-Self (Peng et al., 2025) are drop-in and can be applied to any pretrained MDM, however they lack a precise target: scores are random (DFM, ReMDM), evaluated on a different input than $\mathbf{x}_{t_\ell}$ (ReMDM-conf), or inferred from logits that were not trained to present the token quality (P2-Self).

- Methods that require rehauling MDM design: GIDD (von Rütte et al., 2025), Informed Corrector (Zhao et al., 2024), indeed articulate a valid target for self-correction but achieve it by changing the pretraining design or architecture, requiring full retraining.

- Proxy-$\hat{\mathbf{x}}$ approaches: Token Critic (Gou et al., 2023) and P2-Train (Peng et al., 2025) train an external scorer to approximate the marginal likelihoods of a sampled clean sequence $\hat{\mathbf{x}}$. For a given $\mathbf{x}_{t_\ell}$, it first samples a reconstructed clean sequence $\hat{\mathbf{x}}$ via $f_\theta$ in a single step, then calculates the quality score via the trained module. Since the pretrained MDM has learned the *per-token* unmasking posterior, not the *joint posterior* across positions, $\hat{\mathbf{x}}$ can deviate a lot from a valid sample from the posterior. This method also requires an extra forward pass for scoring.

We provide a comprehensive explanation of the prior work and their notions of per-token quality in Appendix A. *In short*, prior work either lacks a precise target of per-token quality or alters the design of the MDM itself. *In contrast*, PRISM assigns an explicit per-token target to a given sequence $\mathbf{x}_{t_\ell}$ and estimates it in a single forward pass by fine-tuning the pretrained MDM, while still preserving the MDM's ability to compute per-token posteriors.

## 3. PRISM

In this section, we introduce Plug-in Remasking for Inference-time Self-correction of Masked diffusion mod-

els (**PRISM**). We substantiate its theoretical foundation in Section 3.1, present our empirical modeling design in Section 3.2, and explain how PRISM is used during inference in Section 3.3. Recall the definition of per-token quality from Section 2.2: $g_\star \colon (\mathcal{V} \cup \{\mathbf{m}\})^L \to [0,1]^L$,

$$g_\star^i(\mathbf{y}) \approx (\textit{Per-token quality}) = (\textit{how likely } \mathbf{y}^i \textit{ is given } \mathbf{y}).$$

A natural candidate for the per-token quality is the ground-truth likelihood of the token $\mathbf{y}^i$ when $\mathbf{y}^i$ is masked out of $\mathbf{y}$. We denote this masked sequence by $\mathbf{y} \oplus \mathbf{m}_i$, i.e., $(\mathbf{y} \oplus \mathbf{m}_i)^i = \mathbf{m}$ and $(\mathbf{y} \oplus \mathbf{m}_i)^j = \mathbf{y}^j$ for $j \neq i$. Our desired notion of *per-token quality* is the likelihood of a given token conditioned on the rest of the sequence. When $\mathbf{y}^i$ is consistent with the context $\mathbf{y}$, this likelihood should be high; otherwise, it should be low, marking the position as a candidate for remasking. Formally,

$$g_\star^i(\mathbf{y}) := p(\mathbf{x}^i = \mathbf{y}^i \mid \mathbf{y} \oplus \mathbf{m}_i).$$

To clarify, $p(\mathbf{x}^i = \cdot \mid \cdot)$ is the same as the unmasking posterior used for MDM in Section 2.1.

**Discussion on our notion of per-token quality.** Since our per-token quality is derived from the posterior, it can be sensitive to uncertainty, e.g., in $\mathbf{y} = ([\text{I}], \mathbf{m}, [\text{to}], [\text{school}])$, although the fourth token [school] is valid, $g_\star^4(\mathbf{y}) = p(\mathbf{x}^4 = [\text{school}] \mid ([\text{I}], \mathbf{m}, [\text{to}], \mathbf{m}))$ may be low since there are multiple candidates for that position. In such situations, it cannot "hurt" to remask/resample such tokens, as there is high entropy in the true posterior marginal anyway.

More importantly, our notion of per-token quality is effective at detecting *incorrect* tokens in positions with low uncertainty, e.g., in $\mathbf{y} = ([\text{I}], [\text{eat}], [\text{a}], [\text{apple}], [\text{strudel}])$, the third token [a] is ungrammatical given [apple], and $g_\star^3(\mathbf{y})$ is sharply concentrated on other words like [an],[the]; our quality score would correctly identify [a] for remasking and revision.

**Challenge: Unmasking vs. Remasking training.** Our goal is to train (or obtain) a model $g_\phi$ that approximates $g_\star$. This is challenging because $g_\phi$ must take the sequence where the position $i$ is already unmasked, yet it must approximate the posterior defined with the *masked* context $\mathbf{y} \oplus \mathbf{m}_i$. In contrast, an MDM $f_\theta$ predicts from the masked position itself;

$$g_\phi^i(\mathbf{y}) \approx p(\mathbf{x}^i = \mathbf{y}^i \mid \mathbf{y} \oplus \mathbf{m}_i) \in [0,1],$$
$$f_\theta^i(\cdot \mid \mathbf{z}) \approx p(\mathbf{x}^i = \cdot \mid \mathbf{z}) \in \Delta(\mathcal{V}).$$

Although $g_\phi^i(\mathbf{y})$ and the $f_\theta^i(\cdot \mid \mathbf{y} \oplus \mathbf{m}_i)$'s $\mathbf{y}^i$-th coordinate target the same quantity, they condition on different inputs. Consequently, $g_\phi$ cannot be simply trained or efficiently queried as in MDM.

A naive workaround is a distillation-based approach, supervising $g_\phi^i(\mathbf{y})$ with the teacher signal $f_\theta^i(\cdot \mid \mathbf{y} \oplus \mathbf{m}_i)$. This has two drawbacks: First, even in an infinite data and compute regime, we cannot achieve perfect $g_{\phi^\star}$ if the teacher model $f_\theta$ is imperfect. Second, it is data-inefficient: each training pair $(\mathbf{y}, i)$ requires a fresh evaluation of $f_\theta$ on $\mathbf{y} \oplus \mathbf{m}_i$. Consequently, one forward pass of $f$ supervises only a single index, i.e., covering $m$ clean token positions in a sequence requires $m$ passes.

To address this input mismatch, Zhao et al. (2024) leverage a hollow-transformer $g_{\phi'}$, with the property $g_{\phi'}^i(\mathbf{y} \oplus \mathbf{m}_i) \approx p(\mathbf{x}^i = \mathbf{y}^i \mid \mathbf{y} \oplus \mathbf{m}_i)$, enabling the use of the same loss as MDM. However, this requires adopting a new architecture (with a tweak on the attention mechanism) that processes $\mathbf{y}$ while equivalently behaving as if position $i$ were masked, which entails retraining and limiting plug-and-play applicability to pretrained MDMs.

As we will show, PRISM does not possess these limitations: it is plug-and-play with any pretrained MDM $f_\theta$ (no architectural changes), and—even when $f_\theta$ is imperfect, it recovers the desired per-token quality $g_\phi$ in the infinite-data limit.

### 3.1. Theoretical Foundation of PRISM

In this section, we establish the theoretical foundation of PRISM. We begin by recalling the premise behind the MDM training loss before describing how we adapt this premise to give rise to PRISM.

**Marginalization perspective on MDM training.** We rewrite the MDM training loss in (1).

$$\mathcal{L}(\theta) := \mathbb{E}_{\mathbf{x}, \mathbf{z}} \left[ \frac{1}{n} \sum_{i \,:\, \mathbf{z}^i = \mathbf{m}} -\log f_\theta^i(\mathbf{x}^i \mid \mathbf{z}) \right].$$

We now aim to understand why the $\mathcal{L}(\theta)$ has the unmasking posterior as its unique minimizer. First, the expectation over the joint distribution of $(\mathbf{x}, \mathbf{z})$ can be written by first sampling $\mathbf{z}$ and then $\mathbf{x}$ from the induced posterior. For fixed $\mathbf{z}$, the loss only depends on the posterior of $\mathbf{x}^i$ given $\mathbf{z}$, which is exactly the unmasking posterior $p(\mathbf{x}^i = \cdot \mid \mathbf{z})$. Thus, we can rewrite:

$$\mathcal{L}(\theta) = \mathbb{E}_{\mathbf{z}} \left[ \frac{1}{n} \sum_{i \,:\, \mathbf{z}^i = \mathbf{m}} \mathbb{E}_{v \sim p(\mathbf{x}^i = \cdot \mid \mathbf{z})} \left[ -\log f_\theta^i(v \mid \mathbf{z}) \right] \right],$$

and then we observe that the blue-colored term is the cross-entropy between two distributions $f_\theta^i(\cdot \mid \mathbf{z})$ and $p(\mathbf{x}^i = \cdot \mid \mathbf{z})$, which is minimized at $f_{\theta^\star}^i(\cdot \mid \mathbf{z}) = p(\mathbf{x}^i = \cdot \mid \mathbf{z})$ for every $(\mathbf{z}, i)$. Put differently, the masking process of MDM defines the joint distribution over $(\mathbf{x}, \mathbf{z})$ and fixing $\mathbf{z}$ and *marginalizing* over $\mathbf{x}$ yields a cross-entropy between the network output and the desired unmasking posterior. We use this marginalization technique to build PRISM.

---

**Algorithm 1** PRISM fine-tuning from a pretrained MDM

---

1: *Require:* Pretrained MDM backbone $f_\theta$, per-token quality head $g_\theta$, true distribution $p_{\text{data}}$, regularization constant $\lambda > 0$, number of unmasking tokens $k \in \mathbb{N}$.
2: Sample $\mathbf{x} \sim p_{data}$, $n$, and $\mathbf{z}$ from the masking process.
3: # Sample $\mathbf{y}$ from $\mathbf{z}$ with $f_{\text{sg}(\theta)}$
4: Select a subset $\mathcal{S} \subseteq \{i \mid \mathbf{z}^i = \mathbf{m}\}$ with $|\mathcal{S}| = k$.
5: Obtain $\mathbf{y}$ by replacing $\mathbf{z}^i = \mathbf{m}$ to $v^i \sim f^i_{\text{sg}(\theta)}(\cdot \mid \mathbf{z})$ for each $i \in \mathcal{S}$.
6: # Calculate Loss
7: $\mathcal{L}(\theta) \leftarrow \frac{1}{k} \sum_{i \in \mathcal{S}} \text{BCE}(\mathbf{1}[\mathbf{x}^i = \mathbf{y}^i], g^i_\theta(\mathbf{y}))$  ▷ PRISM Loss
8: $\mathcal{L}(\theta) \leftarrow \mathcal{L}(\theta) + \lambda \times \frac{1}{n} \sum_{j:\mathbf{z}^j=\mathbf{m}} - \log f^j_\theta(\mathbf{x}^j \mid \mathbf{z})$.  ▷ Regularization (MDM loss)

---

**PRISM.** Recall our goal for training $g_\phi$ to predict per-token quality, formally defined as $g^i_{\phi^\star}(\mathbf{y}) = p(\mathbf{x}^i = \mathbf{y}^i \mid \mathbf{y} \oplus \mathbf{m}_i)$. Given a pretrained MDM $f_\theta$, PRISM proceeds as follows.

(a) Construct a pair $(\mathbf{x}, \mathbf{z})$ as in MDM training: sample $\mathbf{x} \sim p_{\text{data}}$ and then draw $\mathbf{z}$ by masking.

(b) Sample $\mathbf{y}$: choose a masked index $i$ from $\mathbf{z}$ and replace it with a clean token $\mathbf{y}^i \sim f^i_\theta(\cdot \mid \mathbf{z})$.

Step (a) defines the same joint distribution over $(\mathbf{x}, \mathbf{z})$ as MDM. Step (b) yields a random variable $(\mathbf{y}, i)$ conditioned on $\mathbf{z}$. Although we do not designate a sampling rule for the index $i$, PRISM's guarantee is invariant to its rule, e.g., uniformly random, confidence-based. As a result, PRISM induces a joint distribution over $(\mathbf{x}, \mathbf{z}, (\mathbf{y}, i))$; for illustration, please see Figure 1 (right). Taking expectation over this joint distribution, we define the PRISM loss:

$$\mathcal{L}(\phi) := \mathbb{E}_{\mathbf{x},\mathbf{z},(\mathbf{y},i)}\left[\text{BCE}(\mathbf{1}[\mathbf{x}^i = \mathbf{y}^i], g^i_\phi(\mathbf{y}))\right], \quad (2)$$

where BCE denotes a Binary Cross-Entropy loss $\text{BCE}(b, p) = -b \log p - (1-b) \log(1-p)$ for a binary label $b \in \{0, 1\}$. This loss is the crux of PRISM, whose unique minimizer is the true per-token quality.

---

**Proposition 3.1** (PRISM). *The per-token quality uniquely minimizes the PRISM loss $\mathcal{L}(\phi)$ (2):*

$$g^i_{\phi^\star}(\mathbf{y}) = p(\mathbf{x}^i = \mathbf{y}^i \mid \mathbf{y} \oplus \mathbf{m}_i).$$

---

Before proving Proposition 3.1, we highlight two key advantages that PRISM offers.

**Advantages.** First, the minimizer $g^i_{\phi^\star}(\mathbf{y})$ *does not* depend on the specific choice of $f_\theta$; $f_\theta$ affects the training objective only through the marginalized distribution of $(\mathbf{y}, i)$, not the minimizer. This contrasts with the aforementioned distillation-based approach, in which an imperfect $f_\theta$ will not give rise to a perfect $g_\phi$. Second, PRISM naturally adapts to a pretrained MDM; a single model can share the same architecture and carry two heads, one for the unmasking posterior and one for per-token quality. We detail the modeling choices of PRISM in Section 3.2.

**PRISM: Guarantee.** Proposition 3.1 can be proven within

a few lines. The key mechanism is that the joint distribution $(\mathbf{x}, \mathbf{z}, (\mathbf{y}, i))$ exhibits a simple posterior given $(\mathbf{y}, i)$; $\mathbf{z}$ is deterministically $\mathbf{z} = \mathbf{y} \oplus \mathbf{m}_i$ and each $\mathbf{x}^i$ is drawn from the same unmasking posterior of MDM, i.e., $v \sim p(\mathbf{x}^i = \cdot \mid \mathbf{z})$. Therefore, fixing $(\mathbf{y}, i)$ and then *marginalizing* the binary label $\mathbf{1}[\mathbf{x}^i = \mathbf{y}^i]$ over $(\mathbf{x}, \mathbf{z})$ yields the simple byproduct,

$$q := \mathbb{E}_{v \sim p(\mathbf{x}^i = \cdot \mid \mathbf{y} \oplus \mathbf{m}_i)}\left[\mathbf{1}[v = \mathbf{y}^i]\right] = p(\mathbf{x}^i = \mathbf{y}^i \mid \mathbf{y} \oplus \mathbf{m}_i),$$

the per-token quality. We then expand the binary cross-entropy loss:

$$\mathcal{L}(\phi) = \mathbb{E}_{(\mathbf{y},i)}\left[-q \log(g^i_\phi(\mathbf{y})) - (1-q) \log(1 - g^i_\phi(\mathbf{y}))\right],$$

which is minimized at $g^i_{\phi^\star}(\mathbf{y}) = p(\mathbf{x}^i = \mathbf{y}^i \mid \mathbf{y} \oplus \mathbf{m}_i)$.

### 3.2. Empirical Design Choice of PRISM

In this section, we present our empirical modeling choices that go into the design of PRISM. We first explain the plug-and-play aspect of PRISM, in which we adapt a pretrained MDM and fine-tune it via the PRISM loss.

**Plug-and-Play approach.** Assume a pretrained MDM $f_\theta$ is given, which by design has learned the unmasking posterior. PRISM augments this model with a lightweight auxiliary adapter that reuses the same backbone to produce a size-$L$ tensor. In words, the unmasking posterior and per-token quality share a single backbone and are computed by separate heads.

For a given sequence, we respectively denote the two head outputs as $f_\theta$ and $g_\theta$. Let $\mathbf{h}_\theta$ denote the final hidden state of the MDM backbone. The unmasking posterior $f_\theta$ is obtained by passing $\mathbf{h}_\theta$ through the unmasking head (which originally exists in pretrained MDM) and then applying the softmax. The per-token quality $g_\theta$ is obtained by passing $\mathbf{h}_\theta$ through the attached head and applying the sigmoid. Sigmoid is used to guarantee $g^i_\theta \in [0, 1]$, making it suitable for the binary cross-entropy loss. Thus, with a single $\theta$ and a lightweight adapter, we *jointly model* the unmasking posterior and then the per-token quality: for a partially

---

masked sequence w,

$$g_\theta^i(\mathrm{w}) \approx p(\mathbf{x}^i = \mathrm{w}^i \mid \mathrm{w} \oplus \mathbf{m}_i) \in [0, 1],$$
$$f_\theta^i(\cdot \mid \mathrm{w}) \approx p(\mathbf{x}^i = \cdot \mid \mathrm{w}) \in \Delta(\mathcal{V}).$$

At a high level, the PRISM recipe is simple: train $g_\theta$ using the PRISM loss defined in (2). Consequently, the parameters updated through $g_\theta$ are those of the attached head and the backbone, excluding the unmasking head. As an alternative to fine-tuning the backbone, one can add a LoRA adapter (Hu et al., 2022) into the backbone and train only the adapter parameters.

**Practical interventions.** We introduce two practical interventions that improve PRISM's efficiency. First, to preserve $f_\theta$'s ability to estimate the unmasking posterior while $g_\theta$ is learning toward the per-token quality, we add the MDM training loss (1) as a regularization term.

Second, in the sampling process for constructing PRISM training pairs $(\mathbf{x}, \mathbf{z}, (\mathbf{y}, i))$, we modify step (b). To obtain $\mathbf{y}$, rather than unmasking a single $\mathbf{y}$ from $\mathbf{z}$, we obtain multiple $(\mathbf{y}, i)$ pairs from one $\mathbf{z}$. This produces multiple training pairs $(\mathbf{x}, \mathbf{z}, (\mathbf{y}, i))$ with varying $(\mathbf{y}, i)$ values from a single forward pass of $f_\theta(\cdot \mid \mathbf{z})$, improving data efficiency. Since this procedure of sample construction minimizes $g_\theta$ (not $f_\theta$), we apply stop-gradient to this $f_\theta$. We also provide the pseudocode at Algorithm 1.

Importantly, the theoretical guarantee in Proposition 3.1 still maintains: the true minimizer $g_\theta$ of this modified PRISM loss continues to encode meaningful per-token quality. We defer the precise statement and proof to Appendix B.3. Moreover, we conduct a careful ablation of different design choices on sampling $\mathbf{y}$ in Appendix D.3 and Appendix D.4.

**Practical advantages**. PRISM formulation not only guarantees to minimize the ground truth per-token quality but also has two practical advantages. First, since $f_\theta$ is already pretrained, its hidden states encode useful representations, likely accelerating $g_\theta$'s training on the per-token quality. Second, the targeted per-token quality is a scalar per position, in contrast to the per-position distribution by the unmasking posterior, making it substantially easier to learn. Perhaps surprisingly, we observe that using much less compute compared to MDM pretraining already yields a meaningful learning of the per-token quality (which we detail these results in Section 4), reinforcing the perspective that PRISM is a cheap *fine-tuning* intervention.

### 3.3. Employing PRISM at Inference

In this section, we explain how a model resulting from PRISM is used to perform remasking at inference. The high-level procedure follows Section 2.1; at each inference step, we perform remasking and unmasking simultaneously, using the learned unmasking posterior and per-token quality,

respectively. Assume we are given a fine-tuned MDM $\theta$ (via PRISM loss) and a monotonically decreasing grid of times $t_0 = 1 > \cdots > t_N = 0$ for inference. We initialize $\mathbf{x}_{t_0} = (\mathbf{m}, \ldots, \mathbf{m})$. At each step $t_\ell$, we obtain $\mathbf{x}_{t_{\ell+1}}$ by these following steps.

(a) Choose a subset of masked tokens $\mathcal{S}$ to unmask.

(b) Choose a subset of clean tokens $\mathcal{T}$ to remask with the lowest quality scores $g_\theta(\mathbf{x}_{t_\ell})$.

(c) For each $i \in \mathcal{T}$, remask $\mathbf{x}_{t_\ell}^i$ and for each $j \in \mathcal{S}$, unmask $\mathbf{x}_{t_\ell}^j$ to a clean token sampled from $f_\theta^j(\cdot \mid \mathbf{x}_{t_\ell}) \in \Delta(\mathcal{V})$.

We emphasize that $f_\theta(\cdot \mid \mathbf{x}_{t_\ell})$ and $g_\theta(\mathbf{x}_{t_\ell})$ are obtained from a *single* forward pass of the backbone. Consequently, our method does not add any computational overhead relative to methods without remasking (vanilla MDM inference) or training-free approaches, e.g, ReMDM.

**Design choices.** There are three design choices in this inference procedure: the number of tokens to unmask at each step–$|\mathcal{S}|$, the rule for choosing $\mathcal{S}$, and the number of tokens to remask at each step–$|\mathcal{T}|$. As our focus is on PRISM's remasking strategy, we do not modify the design choices regarding the rule for selecting $\mathcal{S}$ but follow prior seminal approaches. We also keep the rule for selecting $|\mathcal{T}|$ as simple as possible; For unmasking schemes that assign $|\mathcal{S}|$ stochastically, e.g., (Sahoo et al., 2024; Wang et al., 2025; Shi et al., 2024; Gat et al., 2024)), $|\mathcal{S}| \sim \mathrm{Binom}(n, p)$ where $p$ depends on the discretized time steps and $n$ is the number of masked tokens in a given sequence. Here we draw $|\mathcal{T}| \sim \mathrm{Binom}(L - n, \eta)$ for $\eta > 0$, and accordingly set $|\mathcal{S}| \sim |\mathcal{T}| + \mathrm{Binom}(n, p)$ to preserve the total unmasked count's expectation. For schemes that preallocate $|\mathcal{S}|$s uniformly before inference, e.g., (Nie et al., 2025)), we likewise preallocate $|\mathcal{T}|$ uniformly across the time steps.

## 4. Experiments

In this section, we demonstrate that PRISM is a practically efficient and scalable approach across different domains and scales. In Section 4.1, we apply PRISM to a 30M-scale MDM for Sudoku. In Section 4.2, we apply PRISM to a 170M-scale MDM and evaluate its ability for natural language modeling, and in Section 4.3, we apply PRISM to LLaDA-8B-Instruct (Nie et al., 2025) and evaluate it on a Python coding task. In Section 4.4, we conduct a mechanistic analysis on PRISM to further showcase its efficiency. We first outline the pipeline shared across all experiments. **Experiment pipeline.** Starting from a pretrained MDM, we attach a lightweight adapter and fine-tune it with the PRISM loss. We compare against remasking baselines ReMDM (ReMDM-cap/ReMDM-loop) and ReMDM-conf (Wang et al., 2025), re-evaluated using the same backbone after

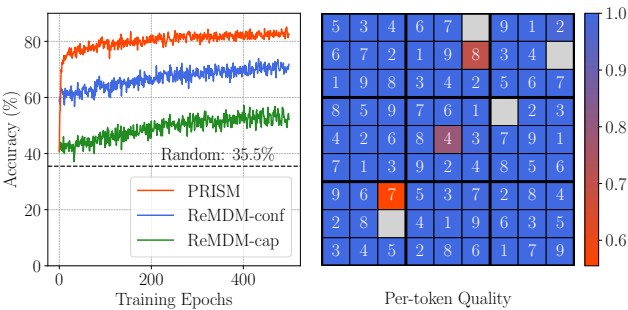

**Figure 2. Left:** PRISM achieves higher accuracy on Sudoku puzzles than baselines. (red curve) **Right:** PRISM detects the incorrect cells by assigning low per-token quality (red-colored cells).

PRISM fine-tuning. We summarize our experimental results as follows:

- **Efficiency**: The PRISM fine-tuned MDM has the ability to self-correct using much fewer samples compared to those used for pretraining.
- **Performance**: The PRISM-tuned MDMs outperform ReMDM and ReMDM-conf, indicating that PRISM learns our desired notion of per-token quality quite well.
- **Operation**: PRISM fine-tuned MDMs calibrate the per-token qualities as desired, and qualitatively correct errors in coding problems.

### 4.1. Motivating Example: Sudoku Puzzle

In this section, we examine PRISM on Sudoku puzzles and show that the learned per-token quality operates as desired. Before moving on to the results, we use Sudoku as a *metaphorical testbed* to explain why ReMDM and ReMDM-conf can fail to capture the intended per-token quality.

**Sudoku as a metaphorical testbed.** ReMDM assigns token quality randomly, which can unnecessarily flag correct cells. ReMDM-conf sets the token quality to the likelihood at the time a cell was unmasked; this is computed on a previous Sudoku board state and can quickly become misaligned from the current board. In contrast, PRISM predicts per-token quality under the current board, thereby reliably identifying incorrect cells.

**Setup.** We construct a corpus of Sudoku puzzles (48k for training and 2k for evaluation), following the setup of (Ben-Hamu et al., 2025; Alp, 2024). Each board is tokenized as a length-89 sequence comprising 81 cell values and 8 [EOL] tokens that separate columns. Following the MDM pretraining recipe of Sahoo et al. (2024), we use a 28.6M Diffusion Transformer (DiT; Peebles & Xie (2023)) architecture as the backbone and pretrain it for 100k iterations ($\approx$ 530 epochs). We then fine-tune it with the PRISM loss, additionally with a lightweight adapter (0.13M parameters).

**Results.** We evaluate the success rate for the same model

across epochs for PRISM and baselines. As shown in Figure 2, PRISM achieves progressively higher success rates during fine-tuning (red curve) and surpasses the baselines. This confirms that the learned per-token quality enables effective self-correction at inference. Notably, PRISM starts to outperform vanilla MDM inference (dashed line) and baselines within only a few epochs, far fewer than the pre-training epochs ($\approx$ 530), corroborating our claim of sample efficiency.

**Visualization of learned per-token quality.** Figure 2 further visualizes PRISM's ability to detect incorrect tokens. On the shown board, three digits are misfilled: $(2, 6) = 8$, $(5, 5) = 4$, and $(7, 3) = 7$. The computed per-token quality assigns low scores to these cells (red), while other cells attain high scores, nearly 0.99 (blue), indicating that the fine-tuned model has almost perfectly learned the per-token quality.

### 4.2. Unconditional Text Generation

**Setup**. We first take a DiT-based MDM with 170M parameters, using a checkpoint from (Sahoo et al., 2024). This MDM is pretrained on the OpenWebText corpus (Gokaslan & Cohen, 2019) with the GPT-2 tokenizer (Radford et al., 2019), a maximum sequence length $L = 1024$, and a total of 262B tokens. We then attach a lightweight 7M parameter adapter, one attention layer applied to the final hidden state, and fine-tune it with the PRISM loss on a total 164M tokens from the same OpenWebText corpus. This fine-tuning is data-efficient, using $\mathbf{1600\times}$ fewer tokens than pretraining. During fine-tuning, we select the subset $\mathcal{S}$ based on confidence scores.

**Results.** We then assess the performance of PRISM and baselines on unconditional generation. Our primary metrics are Generative Perplexity and MAUVE (Pillutla et al., 2021), which respectively assess per-sequence fluency and distributional similarity between generated samples and the reference distribution. Moreover, given the prior observation (Zheng et al., 2024) that a sequence with too many redundant tokens can spuriously yield better, i.e., lower generative perplexity, we report the entropy as a sanity metric. Following standard evaluation protocols (Pillutla et al., 2021; Wang et al., 2025), we generate 5000 samples to compute both metrics, using GPT-2-Large as the reference model. As shown in Figure 3.(a), PRISM outperforms baselines on both metrics (red curves) while achieving a comparable entropy, especially in the regime with fewer sampling steps. See Appendix D.1 for more details.

### 4.3. Applying PRISM to 8B MDM

**Setup.** We take LLaDA-8B-Instruct (Nie et al., 2025), an open-sourced 8B MDM. While freezing the backbone, we add an additional head and a LoRA adapter (Hu et al., 2022),

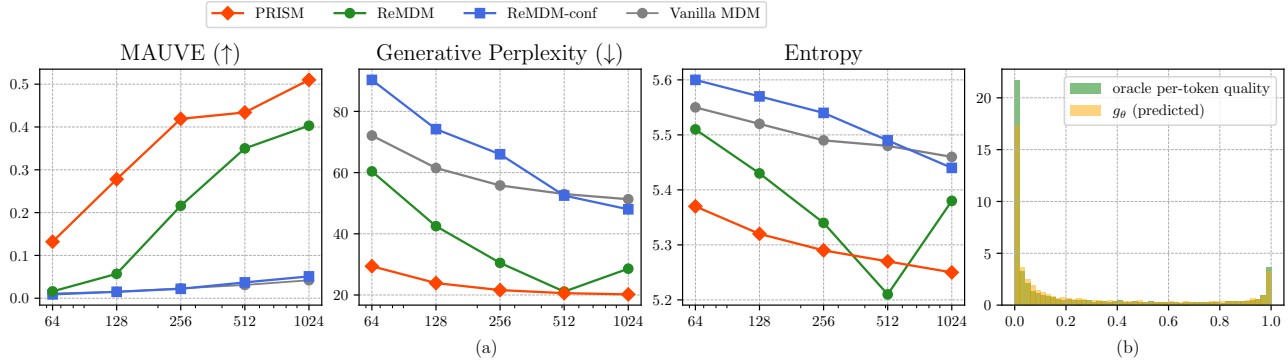

*Figure 3.* **(a) Unconditional text generation performance.** Metrics are evaluated at NFE $\in \{64, 128, 256, 512, 1024\}$; PRISM (red) outperforms baselines (ReMDM: green, ReMDM-conf: blue, Vanilla MDM (without remasking): gray), particularly at lower sampling steps. Detailed numerical results are reported in Table 2. **(b) Calibration study.** PRISM fine-tuned model's per-token quality closely tracks the empirical likelihood.

| | **HumanEval** | | | **MBPP** | | |
|---|---|---|---|---|---|---|
| **Sampling steps** | 256 | 512 | 1024 | 256 | 512 | 1024 |
| PRISM | **28.0** | **39.0** | **42.7** | **21.8** | **29.1** | 32.3 |
| ReMDM | 26.2 | 34.8 | 42.5 | 19.7 | 28.5 | **32.9** |
| ReMDM-conf | 25.6 | 34.1 | 36.6 | 20.1 | 29.0 | 32.5 |
| MDM | 25.6 | 34.1 | 36.6 | 18.2 | 27.8 | 31.9 |

*Table 1.* Zero-shot performance on coding tasks.

for a total of $\approx 250\text{M}$ trainable parameters. We fine-tune it using the PRISM loss on the opc-sft-stage2 (Huang et al., 2024) (0.1M pairs) for 100 epochs, which takes $\approx 30$ hours on 12 NVIDIA H100 GPUs.

**Results.** We evaluate the resulting model in a zero-shot setting on MBPP (Austin et al., 2021b) and HumanEval (Chen, 2021), a challenging Python coding benchmarks. As shown in Table 1, PRISM outperforms ReMDM and ReMDM-conf across different numbers of sampling steps. We detail training and inference configurations in Appendix D.8, D.9.

### 4.4. Mechanistic Analysis of PRISM

In this section, we investigate PRISM's behavior at a *fine-grained level*, showing that the overall PRISM pipeline operates as desired.

**Calibration analysis.** We measure how well the learned per-token quality scores are calibrated to the targeted likelihood. Using the fine-tuned model from Section 4.2, we collect per-token quality scores $g_\theta^i(\mathbf{x}_t)$ for all inference-time sequences $\mathbf{x}_t$s and their clean positions $i$. Then we also compute the corresponding targeted likelihood queried from $f_\theta$, namely $f_\theta^i(\mathbf{x}_t^i \mid \mathbf{x}_t \oplus \mathbf{m}_i)$. Surprisingly, the resulting calibration curves (Figure 3.(b); additional results deferred to Figure 6 in the appendix) closely track the queried likelihoods, indicating that $g_\theta^i(\mathbf{x}_t)$ closely calibrates the target posterior.

**Error-correction analysis.** To assess the actual error-correction behavior of PRISM, we take the model in Sec-

tion 4.3 and measure how often it corrects the syntax errors. Across 164 problems in HumanEval, PRISM corrects 19 out of 26 syntax errors. This provides fine-grained evidence that PRISM can identify and remediate errors in practice. For clarity, we report several failure and success instances in Appendix D.7.

**PRISM-tuned model's knowledge transfers to another.** As we underscore in Section 3.1, the ground-truth per-token quality is defined with respect to the data distribution and thus does not depend on a particular pretrained model. This suggests that a PRISM fine-tuned MDM $A$ can be applied to another pretrained MDM $B$ as a self-corrector, given they operate over the same distribution, e.g., Sudoku puzzles, coding.

As a proof-of-concept experiment on this idea, we apply our PRISM fine-tuned model in Section 4.3 as a self-corrector while using LLaDA-8B-Instruct (not fine-tuned) as a backbone unmasking model. On HumanEval under the same evaluation protocol, self-correction yields a *substantial gain of* $10.9\%$ *(from* $23.2\%$ *to* $34.1\%$*)*. This indicates that the PRISM-tuned model can serve as a general-purpose self-corrector for other pretrained MDMs.

**Learning likelihood itself vs. scalar.** PRISM learns a scalar per-token quality score, rather than the full per-position likelihood $p(\mathbf{x}^i = \cdot \mid \mathbf{y} \oplus \mathbf{m}_i)$. The main motivation is training efficiency: as a scalar score is already informative for self-correction, and learning a full categorical distribution can be computationally inefficient.

The PRISM framework can also be extended to likelihood learning by replacing the BCE objective in Equation 2 with a cross-entropy objective. This richer supervision can distinguish genuinely incorrect tokens from high-entropy positions with many plausible alternatives, enabling uncertainty-aware remasking at additional training cost. We provide additional discussion in Appendix D.6.

# 5. Conclusion

**Discussion on experimental results.** MDMs model per-position unmasking posteriors rather than the joint distribution over positions. Consequently, with few-step inference, they are prone to dependency errors across positions (Liu et al., 2024a; Xu et al., 2024; Kang et al., 2025), which creates a natural opportunity for our self-correction mechanism to detect erroneous predictions. Indeed, the performance gap between PRISM and the vanilla inference increases as the number of sampling steps decreases (Figure 3, Table 1).

**Outlook.** We introduced PRISM, a plug-and-play fine-tuning framework that equips pretrained MDMs with self-correction ability, addressing prevalent limitations of prior work. PRISM operates under a theoretically grounded fine-tuning loss and shows efficiency across multiple domains and scales.

A unique feature of the PRISM loss is that it provably learns per-token quality without fully generating a clean sequence. This sheds light on PRISM's potential compared to other approaches, such as reinforcement learning, where full sequence generation is necessary to obtain a learning signal.

That said, our notion of per-token quality score has limits; as it is based on the per-position posterior marginals, it cannot fully capture global errors, e.g., reasoning. To our knowledge, nevertheless, no prior work principally even targets the per-token quality within the efficient training objective. We therefore view this work as a step that *opens the door* to developing frameworks that enable discrete diffusion models to correct global reasoning errors.

Stepping back, this work is part of a broader program to build generative models that compose discrete sequences the way that humans do–through correction and reordering. The flexibility of MDMs at inference time offers a key stepping stone towards this goal.

# Impact Statement

This paper advances understanding of discrete diffusion models and contributes to the broader machine learning literature. We do not anticipate societal impacts that warrant specific discussion beyond these general considerations.

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

# A. Related Works

## A.1. Background and Related Literature

**Masked Diffusion Models.** Masked Diffusion Models (MDMs) were previously built on continuous-time Markov chains over discrete spaces (Hoogeboom et al., 2021). Subsequently, Austin et al. (2021a) introduced D3PM with several families of transition kernels, followed by SEDD (Lou et al., 2023). These discrete diffusions can also be understood through the lens of discrete flows (Campbell et al., 2024; Gat et al., 2024), which parallel flow matching and stochastic interpolants over continuous spaces (Albergo & Vanden-Eijnden, 2022; Albergo et al., 2023; Lipman et al., 2022). Among transition-kernel designs, the absorbing state (*masking*) kernel has been particularly popular due to its simple formulation and training objective. Subsequent work established a theoretical framework for these models (Zheng et al., 2023; Sahoo et al., 2024; Zheng et al., 2024). Follow-up works have scaled MDMs up to billions of parameters, showcasing their potential compared to autoregressive models (Nie et al., 2025; Ye et al., 2025; DeepMind, 2025; Labs et al., 2025; Gong et al., 2025).

**Inference-time strategies for discrete diffusion models.** A key feature of MDMs is their inference-time flexibility, highlighted by Zheng et al. (2024); Ou et al. (2024) and aligned with our any-order perspective in Section 2.1. Recall the MDM inference procedure from Section 2.1:

(a) Choose a subset of masked positions $\mathcal{S} \subseteq \{i \mid \mathbf{x}_{t_\ell}^i = \mathbf{m}\}$ to unmask.

(b) For each $i \in \mathcal{S}$, unmask by sampling a clean token from $f_\theta^i(\cdot \mid \mathbf{x}_{t_\ell}) \in \Delta(\mathcal{V})$ and setting $\mathbf{x}_{t_\ell}^i$ accordingly.

The choice of $\mathcal{S}$ is entirely flexible, and an appropriate selection can substantially improve downstream task performance. Since our work focuses on self-correction, i.e., re-masking, we only briefly survey prior work on unmasking strategies (Kim et al., 2025; Peng et al., 2025; Chang et al., 2022; Ben-Hamu et al., 2025; Rout et al., 2025). A complementary line of work studies inference-time planning strategies for discrete diffusion with uniform transition kernels (Liu et al., 2024b; Varma et al., 2024); these methods fall outside the scope of MDMs.

## A.2. Details on prior work on remasking strategy (self-correction) for MDM

In this section, we detail our discussion in Section 2.2, focusing on their different notions of per-token qualities. We recall the notion of per-token quality stated in Section 2.2.

$$g_\star \colon (\mathcal{V} \cup \{\mathbf{m}\})^L \to [0,1]^L, \quad g_\star^i(\mathbf{y}) \approx (\text{Per-token quality}) = (\text{"how likely" } \mathbf{y}^i \text{ is given } \mathbf{y}).$$

- **DFM** (Gat et al., 2024) and **ReMDM** (Wang et al., 2025) randomly remask clean tokens in $\mathbf{y}$, which is equivalent to assigning i.i.d. per-token qualities $g_\star^i(\mathbf{y}) \sim \text{Unif}[0,1]$.

- **P2-Self** (Peng et al., 2025) defines per-token quality using the pretrained MDM's score for the observed token under input $\mathbf{y}$, e.g., the probability $f_\theta^i(\mathbf{y}^i \mid \mathbf{y})$. Given that $\mathbf{y}^i \neq \mathbf{m}$, the model was never trained on this index in this state; it only encodes a meaningful quantity (unmasking posterior) when $\mathbf{y}^i = \mathbf{m}$. This notion of per-token quality is therefore not grounded.

- **ReMDM-conf** (Wang et al., 2025) takes the per-token quality to be the likelihood at the step the token was unmasked: if position $i$ was unmasked at $t_k > t_\ell$ (i.e., an earlier step), set $g_\star^i := f_\theta^i(\mathbf{x}_{t_k}^i \mid \mathbf{x}_{t_k}) = f_\theta^i(\mathbf{x}_{t_\ell}^i \mid \mathbf{x}_{t_k})$. As noted, this ignores the current state $\mathbf{x}_{t_\ell}$ and can therefore be misaligned with the model's present beliefs.

- **Token Critic** (Lezama et al., 2022a;b) and **P2-train** (Peng et al., 2025) train an auxiliary scorer for the likelihood of a given prediction of clean sequence $\hat{\mathbf{x}}$. Given $\mathbf{x}_{t_\ell}$, they first form a full completion $\hat{\mathbf{x}}$ by unmasking each mask token at position $i$ from $f_\theta^i(\cdot \mid \mathbf{x}_{t_\ell})$, then evaluate quality by passing this $\hat{\mathbf{x}}$ to the external module. A key limitation is that $\hat{\mathbf{x}}$ is drawn from per-token unmasking posteriors, not the ground-truth posterior across positions; thus, even with a perfect pretrained MDM, $\hat{\mathbf{s}}$ can deviate substantially from the true likelihood $\mathbf{x} \sim p_{\text{data}}$, losing a theoretical guarantee.

- **GIDD** (von Rütte et al., 2025) introduces a family of transition kernels that linearly combine uniform and absorbing (masking) kernels. Models pretrained under this kernel encode posteriors at clean-token positions–however, this requires training under the new kernel and is not directly applicable to an off-the-shelf pretrained MDM, altogether losing the benefit of MDM over other types of kernels.

- **Informed corrector** (Zhao et al., 2024) employs a hollow-transformer $g_{\phi'}$ satisfying $g_{\phi'}^i(\mathbf{y}) = g_{\phi'}^i(\mathbf{y} \oplus \mathbf{m}_i) \approx p(\mathbf{x}^i = \mathbf{y}^i \mid \mathbf{y} \oplus \mathbf{m}_i)$, enabling the same training loss as MDM. This, however, mandates a modified architecture (attention tweak) that processes $\mathbf{y}$ as if position $i$ were masked, necessitating retraining.

In summary, while these methods improve over vanilla MDM inference without remasking, they either substantially reshape the MDM pipeline or rely on imprecise notions and proxies for per-token quality.

## B. Training Algorithms and Guarantees of PRISM

### B.1. PRISM fine-tuning algorithm

### B.2. Marginal Equivalence of the Masking Process

We formalize that the two masking procedures stated in Section 2.1 induce the same conditional distribution of $\mathbf{z}$ given $\mathbf{x}$, hence the same joint distributions $(\mathbf{x}, \mathbf{z})$. It suffices to show they generate the same distribution over masked index sets. Consider (b): draw $t \sim \mathrm{Unif}[0,1]$ and for each $i$, replace $\mathbf{x}^i \in \mathcal{V}$ with $\mathbf{m}$ with probability $t$. The probability of $n$ tokens to be masked is:

$$\int_0^1 \binom{L}{n} t^n (1-t)^{L-n} dt = \binom{L}{n} \mathrm{B}(n+1, L-n+1) = \frac{1}{L+1},$$

where B denotes the Beta function. This coincides with procedure (a), where $n$ is uniformly sampled from $\{0, 1, \ldots, L\}$.

### B.3. PRISM provably learns the per-token quality

In this section, we restate Proposition 3.1 and detail its proof. The proof essentially follows Section 3.1, but we provide the detailed calculations here.

**Proposition 3.1 (restated).** Let the PRISM loss be

$$\mathcal{L}(\phi) = \mathbb{E}_{\mathbf{x}, \mathbf{z}, (\mathbf{y}, i)} \left[ \mathrm{BCE}\big(\mathbf{1}[\mathbf{x}^i = \mathbf{y}^i], g_\phi^i(\mathbf{y})\big) \right], \quad \mathrm{BCE}(b, p) = -b \log p - (1-b) \log(1-p).$$

Then the unique minimizer satisfies, for every $i$ and $\mathbf{y}$,

$$g_{\phi^\star}^i(\mathbf{y}) = p\big(\mathbf{x}^i = \mathbf{y}^i \mid \mathbf{y} \oplus \mathbf{m}_i\big).$$

*Proof.* First, the expectation over the joint distribution $(\mathbf{x}, \mathbf{z}, (\mathbf{y}, i))$ can be written by first sampling $(\mathbf{y}, i)$ and then $(\mathbf{z}, \mathbf{x})$ from the induced posterior. We observe that this induced posterior admits a handy expression:

$$\mathbf{z} = \mathbf{y} \oplus \mathbf{m}_i, \quad v \sim p(\mathbf{x}^i = \cdot \mid \mathbf{z}).$$

Here $p(\mathbf{x}^i = \cdot \mid \cdot)$ is the unmasking posterior given by the MDM's masking process (Section 3.1), and to avoid notational confusion, we introduce a dummy variable $v$. Next, we expand the expectation term.

$$\mathcal{L}(\phi) = \mathbb{E}_{(\mathbf{y}, i)} \left[ \mathbb{E}_{v \sim p(\mathbf{x}^i = \cdot \mid \mathbf{z}), z = \mathbf{y} \oplus \mathbf{m}_i} \left[ \mathrm{BCE}\big(\mathbf{1}[v = \mathbf{y}^i], g_\phi^i(\mathbf{y})\big) \right] \right]$$
$$= \mathbb{E}_{(\mathbf{y}, i)} \left[ -q \log(g_\phi^i(\mathbf{y})) - (1-q) \log(1 - g_\phi^i(\mathbf{y})) \right],$$

where

$$q := \mathbb{E}_{v \sim p(\mathbf{x}^i = \cdot \mid \mathbf{y} \oplus \mathbf{m}_i)} \left[ \mathbf{1}[v = \mathbf{y}^i] \right] = p(\mathbf{x}^i = \mathbf{y}^i \mid \mathbf{y} \oplus \mathbf{m}_i).$$

Finally, we conclude that $g_{\phi^\star}^i(\mathbf{y}) = q = p(\mathbf{x}^i = \mathbf{y}^i \mid \mathbf{y} \oplus \mathbf{m}_i)$ since $f_q(x) = -q \log x - (1-q) \log(1-x)$ is minimized at $x = q \in (0, 1)$. $\square$

### B.4. Extension on PRISM's provable guarantee

In this section, we provide the pseudocode of the practically used PRISM fine-tuning pipeline and then state its theoretical guarantee. We note that the choice of index set $\mathcal{S}$ at line 5 of Algorithm 1 is flexible; it can either be random or confidence-based. To clarify, at line 6, although we use $f_\theta$ to obtain $\mathbf{y}$, we use the stop-gradient operation.

Note that the process of obtaining $\mathbf{y}$ and $\mathcal{S}$ can be interpreted as a process of obtaining multiple $(\mathbf{y}, i)$ pairs for each $i \in \mathcal{S}$. Under this, the expectation over $(\mathbf{x}, \mathbf{z}, (\mathbf{y}, \mathcal{S}))$ is equivalent to the expectation over $(\mathbf{x}, \mathbf{z}, (\mathbf{y}, i))$. Therefore, both formulations of the loss function are valid and equivalent.

$$\mathcal{L}(\theta) = \mathbb{E}_{\mathbf{x}, \mathbf{z}, (\mathbf{y}, \mathcal{S})} \left[ \frac{1}{|\mathcal{S}|} \sum_{i \in \mathcal{S}} \mathrm{BCE}\big(\mathbf{1}[\mathbf{x}^i = \mathbf{y}^i], g_\theta^i(\mathbf{y})\big) \right] = \mathbb{E}_{\mathbf{x}, \mathbf{z}, (\mathbf{y}, i)} \left[ \mathrm{BCE}\big(\mathbf{1}[\mathbf{x}^i = \mathbf{y}^i], g_\theta^i(\mathbf{y})\big) \right].$$

Now we provide the theoretical guarantee on our PRISM loss.

**Proposition B.1** (Provable guarantee of PRISM-extension). *For a fixed $1 \le k \le L$, the PRISM loss*

$$\mathcal{L}(\phi) = \mathbb{E}_{\mathbf{x},\mathbf{z},(\mathbf{y},i)} \left[ \text{BCE}(\, \mathbf{1}[\mathbf{x}^i = \mathbf{y}^i]\,,\, g_\phi^i(\mathbf{y})\,) \right]$$

*has the unique minimizer*

$$g_{\phi^\star}^i(\mathbf{y}) = \mathbb{E}_{\mathcal{S} \sim \Pi(\cdot \,|\, \mathbf{y}, i)} \left[ p(\mathbf{x}^i = \mathbf{y}^i \mid \mathbf{y} \oplus \mathbf{m}_\mathcal{S}) \right],$$

*where the expectation is taken over the posterior distribution over $\mathcal{S}$ given $\mathbf{y}$ and $i$, induced by the joint distribution above. Here $\mathbf{y} \oplus \mathbf{m}_\mathcal{S}$ is a sequence obtained from $\mathbf{y}$ by replacing its $i$-th token with $\mathbf{m}$ for every $i \in \mathcal{S}$, aligning the notation in Section 3.1.*

*Proof.* Before proving this, we clarify some notations. The notation $\mathbf{y} \oplus \mathbf{m}_\mathcal{S}$ extends the notation $\mathbf{y} \oplus \mathbf{m}_i$ from Section 3.1 into the case where we mask multiple tokens, more precisely, $\mathbf{y} \oplus \mathbf{m}_{\{i\}}$.

Also, note that the posterior distribution $\mathcal{S} \sim \Pi(\cdot \mid \mathbf{y}, i)$ is naturally induced from the defined joint distribution. To clarify, this posterior inherits a dependence on the rule that we choose the index set $\mathcal{S}$ during the fine-tuning (and also $f_\theta$). This also satisfies the following property: for a fixed $(\mathbf{y}, i)$, sample $\mathcal{S}$ from this posterior distribution $\Pi(\cdot \mid \mathbf{y}, i)$, and then the desired $\mathbf{x}$'s per-position posterior follows $v \sim p(\mathbf{x}^i = \cdot \mid y \oplus \mathbf{m}_\mathcal{S})$. This property holds since the sampling process of $(\mathbf{x}, \mathbf{z}, (\mathbf{y}, i))$ forms a Markov chain–$\mathbf{z}$ is conditioned on $\mathbf{x}$ and $(\mathbf{y}, i)$ is conditioned on $\mathbf{x}$.

By using this property, we now complete the proof; the remaining parts are the same as Proposition 3.1.

$$\begin{aligned}
\mathcal{L}(\phi) &= \mathbb{E}_{\mathbf{x},\mathbf{z},(\mathbf{y},i)} \left[ \text{BCE}(\, \mathbf{1}[\mathbf{x}^i = \mathbf{y}^i]\,,\, g_\phi^i(\mathbf{y})\,) \right] \\
&= \mathbb{E}_{y,i} \mathbb{E}_{\mathcal{S} \sim \Pi(\cdot|\mathbf{y},i)} \mathbb{E}_{v \sim p(\mathbf{x}^i = \cdot \,|\, \mathbf{z}), \mathbf{z} = \mathbf{y} \oplus \mathbf{m}_\mathcal{S}} \left[ \text{BCE}(\, \mathbf{1}[v = \mathbf{y}^i]\,,\, g_\theta^i(\mathbf{y})) \right] \\
&= \mathbb{E}_{y,i} \mathbb{E}_{\mathcal{S} \sim \Pi(\cdot|\mathbf{y},i)} \left[ -q \log(g_\phi^i(\mathbf{y})) - (1-q) \log(1 - g_\phi^i(\mathbf{y})) \right],
\end{aligned}$$

where $q = p(\mathbf{x}^i = \mathbf{y}^i \mid \mathbf{y} \oplus \mathbf{m}_\mathcal{S})$. Therefore, the expectation over $\mathcal{S} \sim \Pi(\cdot \mid \mathbf{y}, i)$ *absorbs* into $q$, finally derived to our desired form;

$$\begin{aligned}
\mathcal{L}(\phi) &= \mathbb{E}_{\mathbf{x},\mathbf{z},(\mathbf{y},i)} \left[ \text{BCE}(\, \mathbf{1}[\mathbf{x}^i = \mathbf{y}^i]\,,\, g_\phi^i(\mathbf{y})\,) \right] = \mathbb{E}_{y,i} \left[ -q' \log(g_\phi^i(\mathbf{y})) - (1-q') \log(1 - g_\phi^i(\mathbf{y})) \right], \\
q' &= \mathbb{E}_{\mathcal{S} \sim \Pi(\cdot|\mathbf{y},i)} \left[ p(\mathbf{x}^i = \mathbf{y}^i \mid \mathbf{y} \oplus \mathbf{m}_\mathcal{S}) \right].
\end{aligned}$$

$\square$

**Discussion on a new notion of per-token quality.** Proposition B.1 guarantees that the minimizer of the extended PRISM loss satisfies

$$g_{\phi^\star}^i(\mathbf{y}) = \mathbb{E}_{\mathcal{S} \sim \Pi(\cdot|\mathbf{y},i)} \left[ p(\mathbf{x}^i = \mathbf{y}^i \mid \mathbf{y} \oplus \mathbf{m}_\mathcal{S}) \right].$$

Note that by construction, we always have $i \in \mathcal{S}$. This encompasses the $k = 1$ case in Section 3.1; When $k = 1$, the posterior distribution $\Pi(\cdot \mid \mathbf{y}, i)$ collapses to a point mass at $\mathcal{S} = \{i\}$, recovering Proposition 3.1.

We emphasize that this extended definition still captures our *intended* notion of per-token quality—namely, the (posterior) likelihood of $\mathbf{y}^i$ given the rest of the context, here defined under a certain distribution over $\mathbf{y} \oplus \mathbf{m}_\mathcal{S}$. Consequently, a model fine-tuned with the (extended) PRISM loss retains the capacity to identify low-quality tokens, while allowing $k > 1$ to average over multiple plausible masked-context views of $\mathbf{y}$.

$f_\theta$'s effect on PRISM. We show in Proposition 3.1 and Proposition B.1 that, the design of (b), in which we use $f_\theta$ to obtain $\mathbf{y}$, does not directly influence our theoretical results. To clarify, when $k = 1$, the minimizer $g_{\phi^\star}^i(\mathbf{y}) = p(\mathbf{x}^i = \mathbf{y}^i \mid \mathbf{y} \oplus \mathbf{m}_i)$ remains fully independent from $f_\theta$ (or the rule for choosing $i$), and when $k > 1$, the only change is indirect: $f_\theta$ (and the rule for choosing index set $\mathcal{S}$ during PRISM fine-tuning) affects the posterior over masked sets $\Pi(\cdot \mid \mathbf{y}, i)$.

Practically, $f_\theta$ *shapes* the training distribution of samples $(\mathbf{y}, i)$ observed during PRISM fine-tuning, potentially aligning it with those encountered at inference. Since the PRISM loss supervises $\mathbf{y}^i$, which is sampled from $f_\theta$, it is likely to be seen at the inference time (since we also use $f_\theta$ during inference to sample clean tokens). We hypothesize that this alignment improves PRISM fine-tuning, and we revisit this point with an ablation on OpenWebText (Appendix D.4).

## C. Employing PRISM at Inference

In this section, we provide details and pseudocode of inference algorithms that we use for our experiments. We first recall from Section 3.3 how PRISM fine-tuned model $(f_\theta, g_\theta)$ is used at inference: At each step $t_\ell$, we obtain $\mathbf{x}_{t_{\ell+1}}$ through the following steps.

(a) Choose a subset of masked tokens $\mathcal{S}$ to unmask.

(b) Choose a subset of clean tokens $\mathcal{T}$ to remask with the lowest quality scores $g_\theta(\mathbf{x}_{t_\ell})$.

(c) For each $i \in \mathcal{T}$, remask $\mathbf{x}_{t_\ell}^i$ and for each $j \in \mathcal{S}$, unmask $\mathbf{x}_{t_\ell}^j$ to a clean token sampled from $f_\theta^j(\cdot \mid \mathbf{x}_{t_\ell}) \in \Delta(\mathcal{V})$.

We now provide the pseudocode for two different cases of PRISM implementation, depending on how we choose the sizes of the unmasking and remasking sets $|\mathcal{S}|$ and $|\mathcal{T}|$. We begin with the case where we stochastically assign $|\mathcal{S}|$ and $|\mathcal{T}|$ from binomial distributions (Algorithm 2), corresponding to our experiments on OpenWebText (Section 4.2).

---

**Algorithm 2** PRISM inference with assignments from binomial distributions

---

1: *Require:* Fine-tuned MDM with unmasking posterior model $f_\theta$, per-token quality head $g_\theta$, discretized time steps $\{t_\ell\}_{\ell=0}^N$, sampling steps $N$, sequence length $L$, remasking token ratio $\eta > 0$.
2: Initialize $\mathbf{x}_1 = \mathbf{x}_{t_1} \leftarrow (\mathbf{m}, \ldots, \mathbf{m})$
3: **for** $\ell = 0$ to $N - 1$ **do**
4: $\quad$ $K \sim \text{Binom}(|\{i \mid \mathbf{x}_{t_\ell}^i \neq \mathbf{m}\}|, \eta)$ and define $\mathcal{M} \leftarrow \{i \mid \mathbf{x}_{t_\ell}^i = \mathbf{m}\}$
5: $\quad$ **if** $K \leq |\mathcal{M}| < L - K$ and $l \geq l_{on}$ **then**
6: $\quad\quad$ $|\mathcal{T}| \leftarrow K, |\mathcal{S}| \leftarrow \lceil \frac{L}{N_s} \rceil + K$
7: $\quad$ **else**
8: $\quad\quad$ $|\mathcal{T}| \leftarrow 0, |\mathcal{S}| \leftarrow \lceil \frac{L}{N_s} \rceil$
9: $\quad$ **end if**
10: $\quad$ Sample $\mathcal{S} \subset \mathcal{M}$ according to the unmasking rule
11: $\quad$ Sample $\mathcal{T} \subset \{i \mid \mathbf{x}_{t_\ell}^i \neq \mathbf{m}\}$ with low-$|\mathcal{T}|$ indices of $g_\theta^\cdot(x_{t_\ell})$
12: $\quad$ Unmask $\mathbf{x}_{t_\ell}^i$ to $v^i \sim f_\theta^i(\cdot \mid \mathbf{x}_{t_\ell})$ for each $i \in \mathcal{S}$
13: $\quad$ Remask $\mathbf{x}_{t_\ell}^j$ for each $j \in \mathcal{T}$
14: $\quad$ $\mathbf{x}_{t_{\ell+1}} \leftarrow \mathbf{x}_{t_\ell}$
15: **end for**
16: **return** $\mathbf{x}_{t_N} = \mathbf{x}_0$

---

Now, we state the algorithm where we pre-allocate $|\mathcal{S}|$ and $|\mathcal{T}|$ before inference (Algorithm 3) with fixed $K$, corresponding to our experiments on Sudoku 4.1 and LLaDA 4.3.

## D. Experiment Details and Ablation Studies

This section provides additional experimental details.

### D.1. Omitted Results

Here, we present the comprehensive results on OpenWebText in terms of MAUVE score, Generative Perplexity (Gen PPL), and Entropy. The graph in Figure 3 is derived from Table 2. We select the subset $\mathcal{S}$ based on confidence scores during fine-tuning.

*Table 2.* Evaluation result for PRISM/baselines across $N$. The best MAUVE scores are **bolded**. Baseline results marked with [†] are taken from (Wang et al., 2025).

| Method | MAUVE (↑) | | | | | Gen PPL. (↓) | | | | | Entropy (↑) | | | | |
|---|---|---|---|---|---|---|---|---|---|---|---|---|---|---|---|
| | *64* | *128* | *256* | *512* | *1024* | *64* | *128* | *256* | *512* | *1024* | *64* | *128* | *256* | *512* | *1024* |
| MDLM[†] | 0.011 | 0.015 | 0.023 | 0.031 | 0.042 | 72.1 | 61.5 | 55.8 | 53.0 | 51.3 | 5.55 | 5.52 | 5.49 | 5.48 | 5.46 |
| ReMDM-conf | 0.009 | 0.015 | 0.022 | 0.037 | 0.051 | 90.3 | 74.2 | 66.0 | 52.5 | 48.0 | 5.60 | 5.57 | 5.54 | 5.49 | 5.44 |
| ReMDM[†] | 0.016 | 0.057 | 0.216 | 0.350 | 0.403 | 60.4 | 42.5 | 30.5 | 21.1 | 28.6 | 5.51 | 5.43 | 5.34 | 5.21 | 5.38 |
| PRISM (ours) | **0.132** | **0.278** | **0.419** | **0.434** | **0.510** | 29.4 | 23.9 | 21.6 | 20.6 | 20.2 | 5.37 | 5.32 | 5.29 | 5.27 | 5.25 |

---

**Algorithm 3** PRISM inference with fixed $K$

---

1: *Require:* Fine-tuned MDM with unmasking posterior model $f_\theta$, per-token quality head $g_\theta$, sampling steps $N$, sequence length $L$, discretized time steps $\{t_\ell\}_{\ell=0}^N$, number of remasking tokens $K$
2: Initialize $\mathbf{x}_1 = \mathbf{x}_{t_1} \leftarrow (\mathbf{m}, \dots, \mathbf{m})$
3: **for** $l = 0$ to $N - 1$ **do**
4:   Define $\mathcal{M} \leftarrow \{i \mid \mathbf{x}_{t_l} = \mathbf{m}\}$
5:   **if** $K \leq |\mathcal{M}| < L - K$ and $l \geq l_{on}$ **then**
6:     $|\mathcal{T}| = K, |\mathcal{S}| = \lceil \frac{L}{N} \rceil + K$
7:   **else**
8:     $|\mathcal{T}| = 0, |\mathcal{S}| = \lceil \frac{L}{N} \rceil$
9:   **end if**
10:   Sample $\mathcal{S} \subset \mathcal{M}$ according to the unmasking rule
11:   Sample $\mathcal{T} \subset \{i \mid \mathbf{x}_{t_\ell}^i \neq \mathbf{m}\}$ with low-$|\mathcal{T}|$ indices of $g_\theta^{\cdot}(x_{t_\ell})$
12:   Unmask $\mathbf{x}_{t_\ell}^i$ to $v^i \sim f_\theta^i(\cdot \mid \mathbf{x}_{t_\ell})$ for each $i \in \mathcal{S}$
13:   Remask $\mathbf{x}_{t_\ell}^j$ for each $j \in \mathcal{T}$
14:   $\mathbf{x}_{t_{\ell+1}} \leftarrow \mathbf{x}_{t_\ell}$
15: **end for**
16: **return** $\mathbf{x}_{t_N} = \mathbf{x}_0$

---

## D.2. Details on the Remasking Strategy for OpenWebText

This section provides a detailed elaboration of the sampling strategies employed for the unconditional text generation in Section 4.2. The main motivation for these interventions is to preserve sentence diversity, which is crucial for evaluated metrics.

- **Staged Remasking**: We restrict the self-correction mechanism to be activated only after a specific step $\ell_{on}$. This restriction is motivated by the observation that low per-token quality scores in the initial sampling phase ($t > t_{\ell_{on}}$) are likely due to a lack of suffix context rather than token-level inaccuracies. Activating the correction mechanism only after a sufficient context prevents spurious modifications to an incomplete sequence.

- $\eta$ **Inverse Scheduling**: Fixing $\eta$ was found to degrade the diversity as the $N$ increases. To mitigate this, an inverse scheduling rule was adopted, dynamically adjusting $\eta$ based on $N$.

To avoid complicated hyperparameter designs, we set these hyperparameters to be a function of $N$. The remasking ratio and the remasking activation step are set to $\eta = \frac{2.56}{N}$ and $l_{on} = \lceil 0.5N \rceil$, respectively.

## D.3. Ablation on Fine-tuning Hyperparameters-$(k, n_y)$

In addition to $k$ (the number of tokens updated simultaneously to sample $\mathbf{y}$ from $\mathbf{z}$), we introduce a second hyperparameter, $n_y \in \mathbb{N}$, to improve data efficiency during the adapter's fine-tuning. As discussed in Section 3.2, we can generate multiple distinct target sequences $\mathbf{y}$ from a single source sequence $\mathbf{z}$ by using different unmasking index sets $\mathcal{S}$. Therefore, $n_y$ denotes the number of unique unmasking sets applied per $\mathbf{z}$ in a single forward pass.

We conducted an ablation study to analyze how the interplay between $k$ and $n_y$ affects the generation performance. We tested variants of $(k, n_y) \in \{(4, 8), (8, 4), (16, 2), (32, 1)\}$, thus remaining the total number of token updates per batch constant ($k \times n_y = 32$) for a fair comparison. All other configurations follow Table 5, except that we set nucleus $p = 0.9$ to sample $\mathbf{y}$ from $\mathbf{z}$, and select $\mathcal{S}$ randomly during PRISM fine-tuning.

Our primary finding is that training the adapter with a smaller number of updating tokens (low $k$) results in a model that performs better at inference time, as measured by the MAUVE score. This can be attributed to a train-test distribution mismatch induced by large $k$. Specifically, updating many tokens simultaneously is likely to introduce joint dependency errors, increasing the chance of sampling a *misleading* sequence $\mathbf{y}$. This forces the adapter to train on a corrupted data distribution that is rarely encountered during inference time. Consequently, this mismatch leads to a poorly calibrated quality estimator, degrading the model's ability to perform effective self-correction.

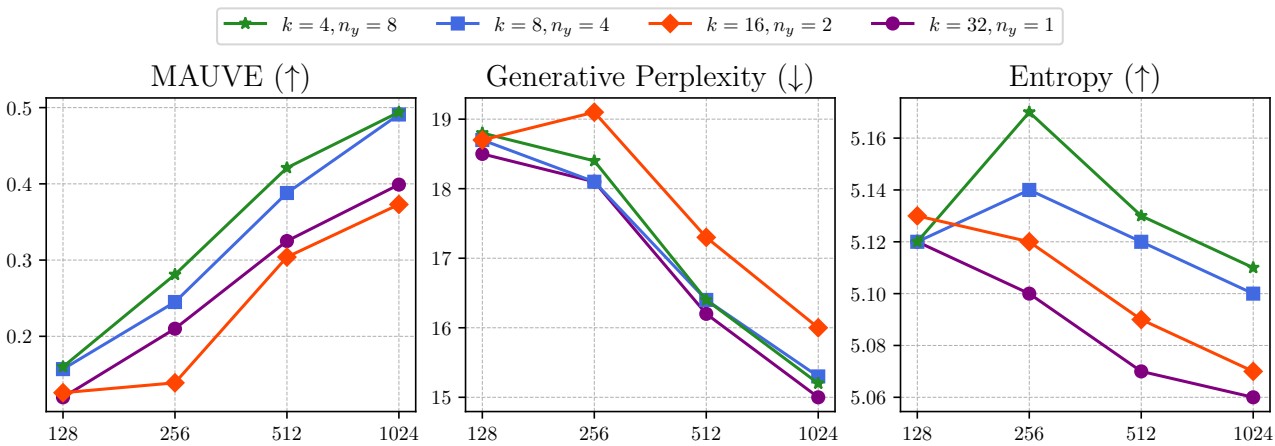

*Figure 4.* Ablation study on fine-tuning hyperparameters $k$ and $n_y$ while holding their product constant ($k \times n_y = 32$). Evaluations are conducted in $(128, 256, 512, 1024)$ sampling steps.

### D.4. Ablation on Fine-tuning Hyperparameters-Nucleus sampling

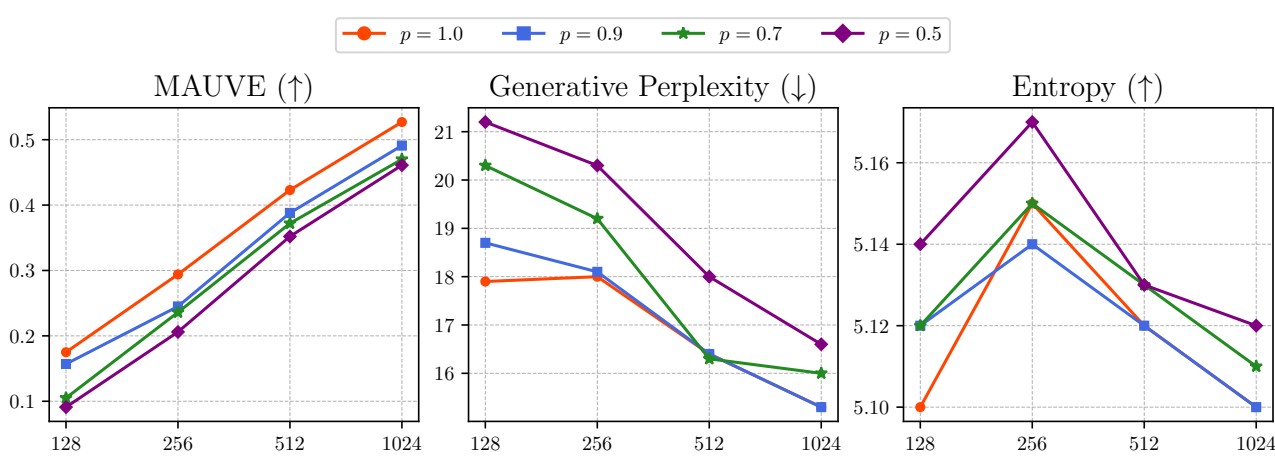

*Figure 5.* Ablation study on nucleus sampling $p$ used to generate target sequences during adapter fine-tuning. Evaluations are conducted in $(128, 256, 512, 1024)$ sampling steps.

We further investigate how the sampling distribution used to generate target sequences $\mathbf{y}$ affects the adapter's fine-tuning process. Specifically, we analyze the impact of nucleus sampling probability $p$ that we use to sample a token $\mathbf{y}^i$ from $f_\theta^i(\cdot \mid \mathbf{z})$, while selecting $\mathcal{S}$ randomly. Our hypothesis is that a less restrictive sampling distribution (large $p$) is more beneficial for training the quality estimator than a shrunk one (small $p$).

Intuitively, a large $p$ induces a relatively diverse sample $\mathbf{y}$. These sequences include not only high-probability tokens but also rare yet plausible alternatives. Training on these plausible-but-imperfect examples reduces exposure bias by forcing the model to calibrate its confidence across a wider range of outcomes, rather than only learning from its own single best predictions. This process cultivates a more robust quality estimator capable of discerning subtle inaccuracies. While this approach may introduce gradients with larger variance, it provides superior coverage of the complex distributions encountered during inference. This ultimately leads to a more effective self-correction mechanism and improved text quality, as validated by the results in Fig. 5.

### D.5. Quantitative Analysis of PRISM: Calibration Study

We provide the calibration analysis result of PRISM in Figure 6.

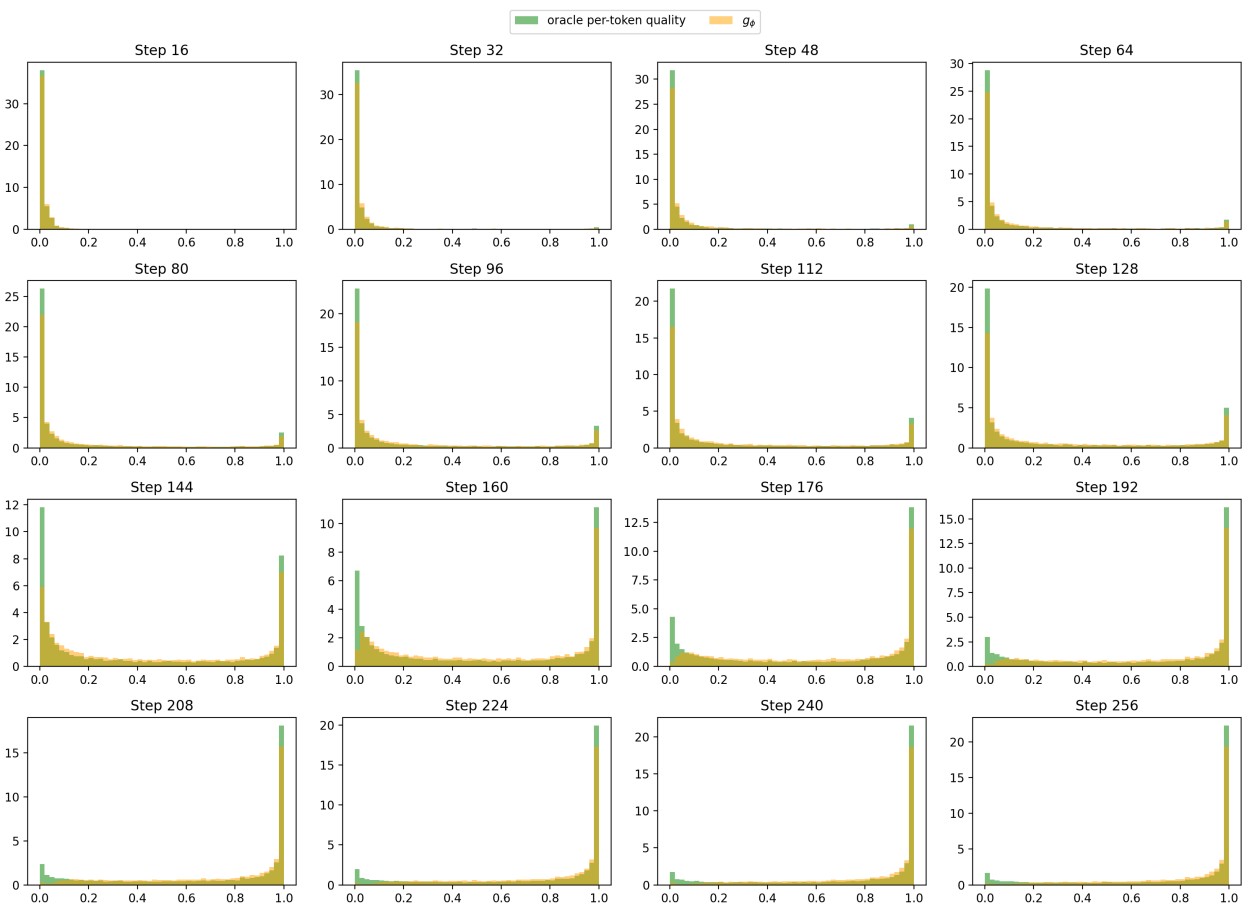

*Figure 6.* Calibration of PRISM per-token quality on OpenWebText during generation without remasking with 256 steps. For each timestep, tokens are binned by predicted quality $g_\theta^i(\mathbf{x}_t)$, and for each bin we plot the empirical probability that token $i$ matches the ground truth, estimated via $f_\theta(\cdot \mid \mathbf{x}_t \oplus \mathbf{m}_i)$, against the bin mean.

## D.6. Learning likelihood instead of scalar

*Table 3.* Comparison between scalar-quality learning and likelihood-learning variants on OpenWebText. PRISM learns a scalar per-token quality score, whereas likelihood-based variants learn the full per-position conditional likelihood. Under the same 10k-step training budget, scalar-quality PRISM is more training-efficient and achieves better MAUVE score. With longer training, likelihood learning becomes competitive, and entropy-aware remasking further improves MAUVE across decoding budgets.

| Method | MAUVE (↑) | | | | | Gen PPL. (↓) | | | | | Entropy (↑) | | | | |
|---|---|---|---|---|---|---|---|---|---|---|---|---|---|---|---|
| | *64* | *128* | *256* | *512* | *1024* | *64* | *128* | *256* | *512* | *1024* | *64* | *128* | *256* | *512* | *1024* |
| PRISM (scalar, 10k) | 0.132 | 0.278 | 0.419 | 0.434 | 0.510 | 29.4 | 23.9 | 21.6 | 20.6 | 20.2 | 5.37 | 5.32 | 5.29 | 5.27 | 5.25 |
| Likelihood (10k) | 0.083 | 0.141 | 0.179 | 0.214 | 0.185 | 18.8 | 15.5 | 14.1 | 13.5 | 13.1 | 5.18 | 5.13 | 5.10 | 5.08 | 5.07 |
| Likelihood (20k) | 0.144 | 0.249 | 0.359 | 0.435 | 0.421 | 18.6 | 15.2 | 13.7 | 13.0 | 12.7 | 5.20 | 5.15 | 5.11 | 5.09 | 5.08 |
| Likelihood + Entropy (20k) | **0.195** | **0.322** | **0.433** | **0.442** | **0.520** | 20.7 | 18.4 | 17.5 | 17.1 | 16.9 | 5.28 | 5.25 | 5.22 | 5.21 | 5.20 |

Our formulation of PRISM learns a scalar per-token quality score $g_\star^i(\mathbf{y})$. This scalar is sufficient for identifying low-quality tokens while remaining lightweight and efficient to train. A natural extension is to learn the full per-position likelihood instead:

$$p(\mathbf{x}^i = \cdot \mid \mathbf{y} \oplus \mathbf{m}_i) \in \triangle(\mathcal{V}).$$

In this section, we study this likelihood-learning variant and analyze the trade-off it introduces for self-correction and remasking. To keep notation consistent with the main text, we slightly overload $g_\phi^i$: in this subsection, $g_\phi^i(\cdot \mid \mathbf{y} \oplus \mathbf{m}_i)$ denotes a distribution-valued likelihood head, whereas in the main body $g_\phi^i(\mathbf{y})$ denotes a scalar per-token quality score.

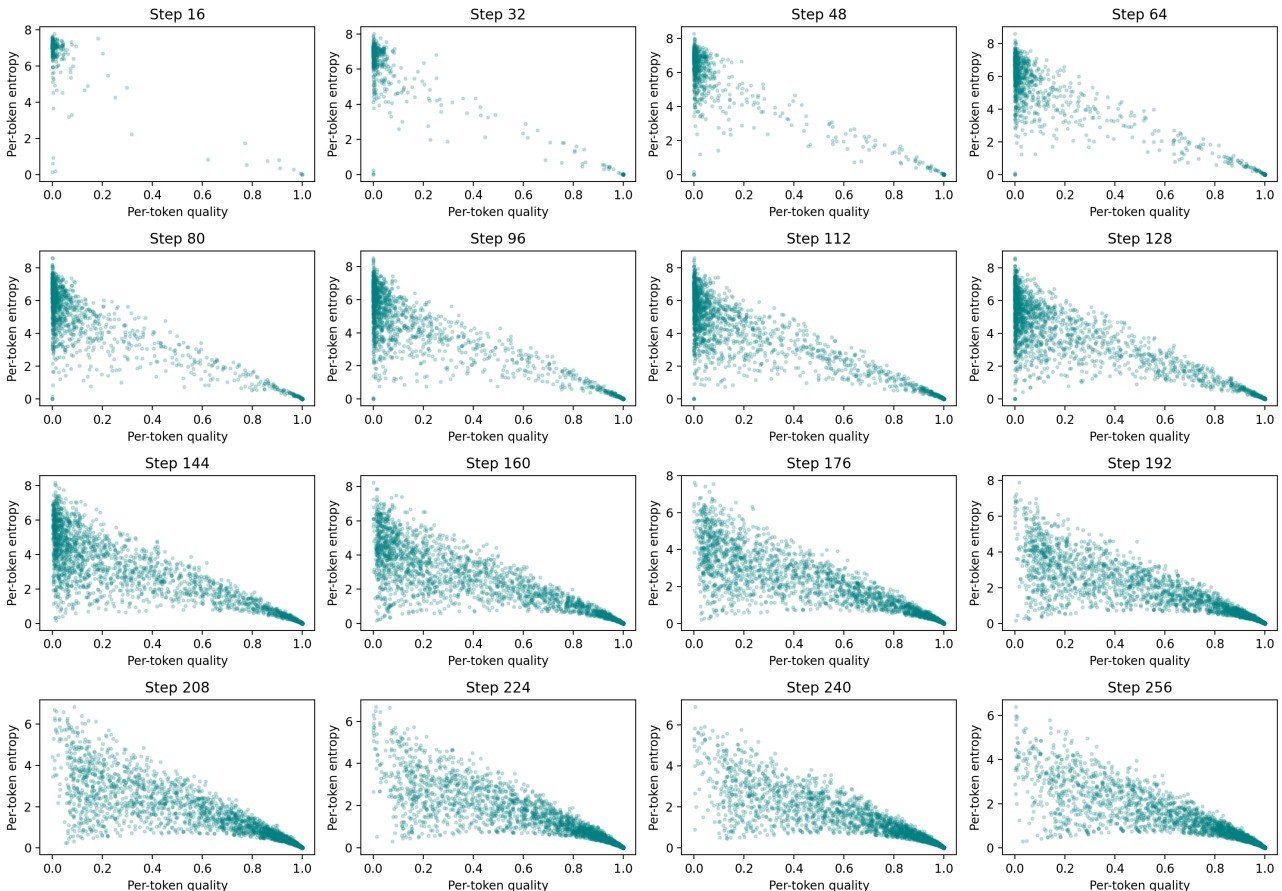

*Figure 7.* Relationship between per-token quality and per-position entropy during OpenWebText generation. Each point corresponds to a revealed token during inference. In this setting, remasking is activated only during the 128–256 step interval. Across denoising steps, many tokens exhibit low quality but high entropy, indicating positions where the current token may be plausible despite receiving a low scalar-quality score.

**Likelihood Learning.** The theoretical framework of PRISM extends directly to likelihood learning. Specifically, replacing the BCE objective in Equation 2 with a cross-entropy objective gives

$$\mathcal{L}_{CE}(\phi) := \mathbb{E}_{\mathbf{x}, \mathbf{z}, (\mathbf{y}, i)} \left[ -\log g_\phi^i(\mathbf{x}^i \mid \mathbf{y} \oplus \mathbf{m}_i) \right], \tag{3}$$

whose minimizer is the conditional likelihood $p(\mathbf{x}_i = \cdot \mid \mathbf{y} \oplus \mathbf{m}_i)$. Unlike PRISM, this variant does not require an additional scalar-quality head. Instead, we can fine-tune the MDM backbone using the existing unmasking head, replacing $g_\phi$ in Equation 3 with $f_\theta$. Compared to scalar-quality learning, likelihood learning provides richer information at each position, but it is also more expensive to learn as it requires learning a full vocabulary distribution rather than a single scalar.

To study this trade-off, we train likelihood-based variants under the same OpenWebText setup used in our main experiments, but with the likelihood objective above and without an additional scalar-quality head. Table 3 shows that likelihood learning is effective, but underperforms scalar-quality PRISM under the same 10k-step training budget. With longer training, however, the likelihood-based variant becomes competitive with scalar-quality PRISM, indicating that the likelihood objective is compatible with the PRISM framework but requires more optimization.

**Richer supervision from learned likelihoods.** A key advantage of likelihood learning is that it exposes uncertainty information through the per-position entropy

$$H_\theta^i(\mathbf{x}_t) := -\sum_{v \in \mathcal{V}} g_\theta^i(\mathbf{x}_t)[v] \log g_\theta^i(\mathbf{x}_t)[v],$$

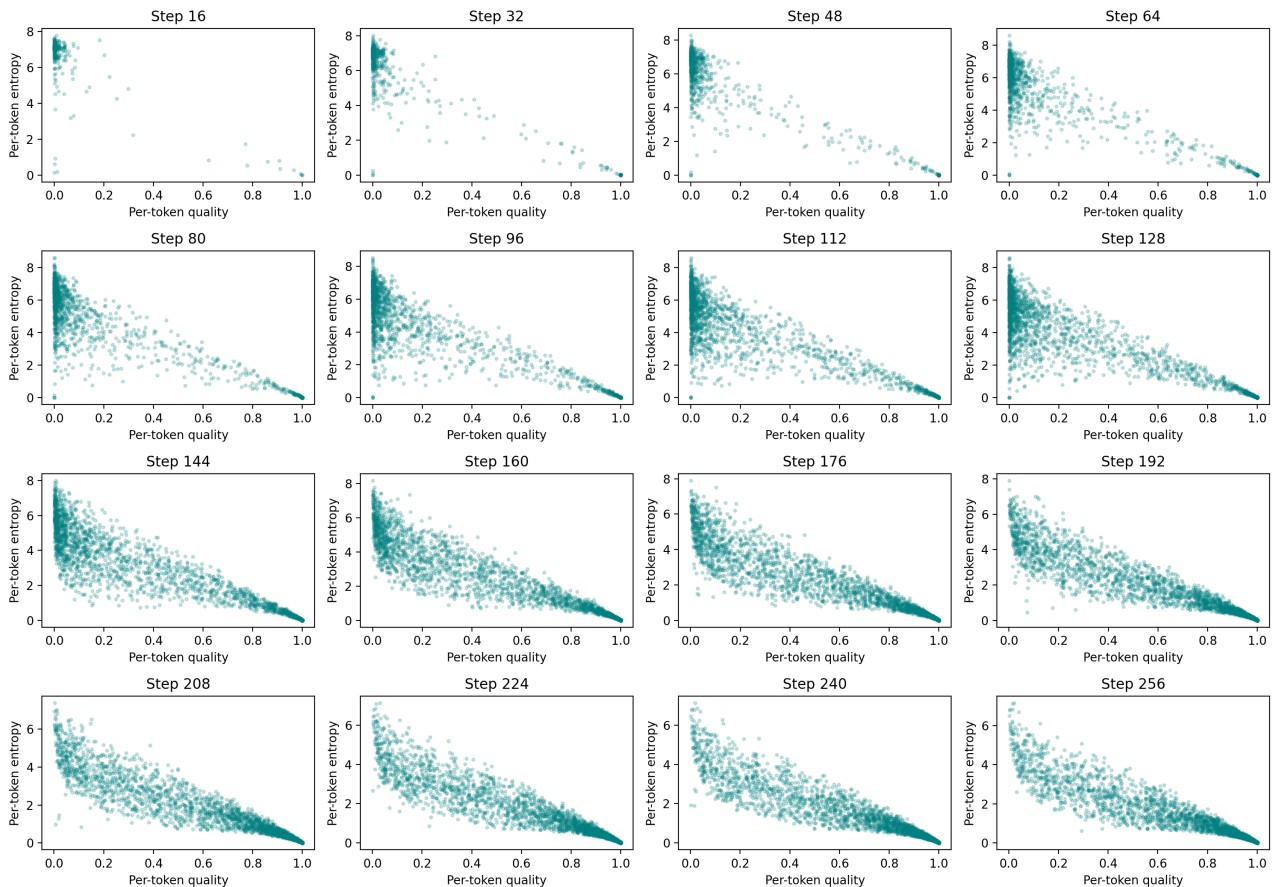

*Figure 8.* Effect of entropy-aware remasking during OpenWebText generation. In this setting, remasking is activated only during the 128–256 step interval. Using the entropy-adjusted score $s^i(\mathbf{x}_t)$ biases remasking toward low-quality, low-entropy tokens, which are more likely to correspond to genuine errors.

This entropy captures the uncertainty of the token distribution at a revealed position $i$. Low per-token quality can arise from two qualitatively different situations: (1) the revealed token is genuinely erroneous, or (2) the token is semantically valid but belongs to a high-entropy position with many plausible alternatives. Remasking tokens in the second case can be redundant or even harmful. Figure 7 shows that low-quality, high-entropy tokens appear throughout the denoising process, suggesting that scalar quality alone does not distinguish these two cases.

Motivated by this observation, we consider the following entropy-aware remasking score:

$$s^i(\mathbf{x}_t) = g_\theta^i(\mathbf{x}_t)[\mathbf{x}_t^i] \cdot \exp(H_\theta^i(\mathbf{x}_t)) = \exp\left[\log g_\theta^i(\mathbf{x}_t)[\mathbf{x}_t^i] - \mathbb{E}_v\left[\log g_\theta^i(\mathbf{x}_t)[v]\right]\right],$$

This score compensates for high-entropy positions and therefore avoids unnecessarily remasking tokens that have many plausible alternatives. Using the same remasking schedule with $s^i(x_t)$, we observe that low-quality, low-entropy positions are more selectively remasked, as shown in Figure 8. Quantitatively, the entropy-aware score improves generation quality over the likelihood-only score in Table 3, with especially large gains in the low-sampling-step regime.

That said, this richer supervision comes with a training-efficiency trade-off. On OpenWebText, we observe an approximately $2\times$ training-efficiency gap between scalar-quality learning and likelihood learning. It remains an open question how this gap evolves at larger scales, where learning a full likelihood may become more costly.

### D.7. Quantitative analysis of PRISM: Error correction study

**Case-by-case analysis.** We provide a qualitative breakdown of PRISM's self-correction behavior across representative HumanEval tasks. The code snippets are given in Table 4.

- Syntax error correction (task id 56): The baseline model produces a syntactically invalid program due to an incomplete `elif` statement, which results in a `SyntaxError`. PRISM successfully detects this low-quality token and replaces the invalid branch with a valid `else` clause, yielding a syntactically correct and functionally valid implementation.

- Logical error correction (task id 5): The baseline implementation incorrectly appends the delimiter after every element, including the last one. PRISM revises the code to conditionally insert the delimiter only between elements, correcting the logical structure of the program while preserving the intended functionality.

- (Failure case) overcorrection (task id 25): Although the baseline solution is correct, PRISM rewrites the code into a more complex form that relies on repeated `min` and `max` operations. This transformation introduces an error when the input list becomes empty, illustrating a failure mode where PRISM overcorrects a correct but non-canonical solution.

- (Failure case) global reasoning error (task id 17): Both the baseline model and PRISM fail on this task. The core issue lies in the iteration structure: the program should iterate over multi-character musical symbols rather than individual characters. This type of global structural error is not captured by PRISM's token-level quality estimation, highlighting a limitation of local self-correction.

### D.8. Experiment Configurations

In Table 5, we list the hyperparameter configurations for Sudoku, OWT, and LLaDA experiments.

### D.9. Additional experimental details on LLaDA

Although Table 5 lists the basic configurations for our LLaDA experiments, we provide comprehensive details below.

**Attaching the adapter.** As discussed in Section 4.3, we freeze the LLaDA-8B-Instruct backbone and add (i) an auxiliary head to model per-token quality and (ii) a LoRA adapter on the qkv attention projections (rank 256, dropout ratio 0.1). This yields roughly 250M trainable parameters in total.

**Dataset construction.** For PRISM fine-tuning, we adopt the opc-sft-stage-2 (Huang et al., 2024), comprising $\approx 0.1$M challenging Python coding problems. Each example includes instructions and solutions. Thus, in the masking process, we do *not* mask instruction tokens.

**Training configuration.** We use AdamW (Loshchilov & Hutter, 2017) (learning rate $1.0 \times 10^{-4}$, weight decay 0.01, warmup ratio 0.05) and use a cosine schedule with 5 cycles. Training for 100 epochs takes approximately 30 hours on 12 H100 GPUs.

**Inference details.** For unmasking, we follow Nie et al. (2025) and employ semi-autoregressive inference: the sequence is partitioned into multiple blocks, and decoding proceeds from the leftmost block. We set the nucleus parameter to $p = 0.0$. These two choices are known to be important for competitive coding performance.

For re-masking, across all baselines, we re-mask exactly $K_{\text{block}}$ tokens per block and schedule these re-masking operations at intermediate stages of block-wise inference (we observe degraded performance if re-masking is concentrated only at the beginning or end of a block). For a fair comparison, we report the best performance for PRISM and baselines (ReMDM, ReMDM-conf) over a sweep of $K_{\text{block}}$ at a fixed number of inference steps $N$: specifically, $K_{\text{block}} \in \{4, 6\}$ for $N = 256$, (this is because the number of unmasking steps equals 8 for $N = 256$) and $K_{\text{block}} \in \{8, 10, 12, 14, 16\}$ for $N = 512$ and $N = 1024$.

| | Baseline | PRISM |
|---|---|---|
| Syntax error correction (task 56) | ```python
def correct_bracketing(brackets):
    stack = []
    for bracket in brackets:
        if bracket == <:
            stack.append(bracket)
        elif:   # SyntaxError
            if not stack:
                return False
            stack.pop()
    return not stack
``` | ```python
def correct_bracketing(brackets):
    stack = []
    for bracket in brackets:
        if bracket == <:
            stack.append(bracket)
        else:
            if len(stack) == 0:
                return False
            stack.pop()
    return len(stack) == 0
``` |
| Logical error correction (task 5) | ```python
def intersperse(numbers,
    delimeter):
    result = []
    for num in numbers:
        result.append(num)
        result.append(delimeter)
    return result
``` | ```python
def intersperse(numbers,
    delimeter):
    result = []
    for num in numbers:
        if not result:
            result.append(num)
        else:
            result.append(
                delimeter)
            result.append(num)
    return result
``` |
| Failure case (task 25) | ```python
def strange_sort_list(nums):
    if not nums:
        return []
    nums.sort()
    result = []
    while nums:
        result.append(nums.pop(0)
            )
        if nums:
            result.append(nums.
                pop(-1))
    return result
``` | ```python
def strange_sort_list(nums):
    result = []
    while nums:
        min_val = min(nums)
        result.append(min_val)
        nums.remove(min_val)
        max_val = max(nums)
        result.append(max_val)
        nums.remove(max_val)
    return result
``` |
| Global reasoning failure (task 17) | ```python
def parse_music(s):
    out = []
    for note in s:
        if note == 'o':
            out.append(4)
        elif note == 'o|':
            out.append(2)
        elif note == '.|':
            out.append(1)
    return out
``` | ```python
def parse_music(s):
    out = []
    for note in s:
        if note == 'o':
            out.append(4)
        elif note == 'o|':
            out.append(2)
        elif note == '.|':
            out.append(1)
    return out
``` |

*Table 4.* Qualitative comparison of self-correction behavior.

*Table 5.* Hyperparameter configurations for Sudoku, OWT, and LLaDA.

| Hyperparameter | Sudoku | OWT | LLaDA |
|---|---|---|---|
| **Base Model** | | | |
| Architecture | DiT | DiT | (non-causual) Transfomer |
| Parameters | 28.6M | 169M | 8B |
| Tokenizer | N/A | GPT-2 | LLaDA-8B-Instruct |
| Sequence Length | 81 | 1024 | 4096 |
| **Fine-tuning** | | | |
| Adapter Architecture | linear | attention+linear | projection+linear |
| Adapter Parameters | 133K | 7.9M | 240M |
| Optimizer | AdamW | AdamW | AdamW |
| Learning Rate | $3.0 \times 10^{-4}$ | $1.5 \times 10^{-4}$ | $1.0 \times 10^{-4}$ |
| Weight Decay | 0.0 | 0.0 | 0.1 |
| Gloabl Batch Size | 256 | 256 | 144 |
| $k$ | 4 | 32 | 8 |
| $n_y$ | 1 | 1 | 1 |
| Nucleus $p$ | 1.0 | 1.0 | 0.0 |
| $\lambda$ | 5.0 | 0.5 | 0.5 |
| **Sampling** | | | |
| unmasking strategy | random | random | semi-autoregressive |
| $n_{\text{blocks}}$ | 1 | 1 | 32 |
| Nucleus $p$ | 1.0 | 0.9 | 0.0 |
| Sampling Precision | float64 | float64 | float32 |
| $K$ | 4 | N/A | – |
| $l_{on}$ | 0 | $\lceil 0.5N \rceil$ | – |
| $\eta$ | N/A | $\frac{2.56}{N}$ | N/A |

