# OpenReview forum: "Fine-Tuning Masked Diffusion for Provable Self-Correction"
_ICML.cc/2026/Conference — ICML 2026 regular_

### Official Review · Reviewer_yKha · 2026-03-12

**Soundness:** 2
**Presentation:** 2
**Significance:** 3
**Originality:** 2
**Overall Recommendation:** 2
**Confidence:** 4

**Summary:**

This paper proposes PRISM, a lightweight self-correction approach applied to pretrained MDMs. Specifically, PRISM introduces an additional module that predicts the per-token quality score, which is used to remask possibly incorrect tokens to enable the self-correction generation of MDMs. Empirically, PRISM improves the generation quality of the 170M MDM on Sudoku, and the 8B LLaDA model on code generation, compared with baseline methods.

**Compliance With Llm Reviewing Policy:**

Affirmed.

**Ethical Review Concerns:**

I think this paper (or at least a non-negligible part) is likely to be generated by AI, as the manuscript lacks logical coherence, with inconsistent context and abrupt connections between ideas and claims. Please check it carefully.

**Ethical Review Flag:**

Flag this paper for an ethics review.

**Ethics Expertise Needed:**

["Other Expertise"]

**Key Questions For Authors:**

1. What is the input of the quality head $g_\theta$? Is it a token sequence or a hidden state tensor? According to Algorithm 1, it seems to take a token sequence as the input. However, according to paragraph 3 in Section 3.2, $g_\theta$ takes the hidden state as input. Could the author provide a clear explanation? How does it share parameters with the backbone?

1. Following the first question, according to paragraph 3 in Section 3.2, $f_\theta$ is the unmasking head. Why is it denoted as "Pretrained MDM backbone" in Algorithm 1?

1. Following the first question, according to paragraph 3 in Section 3.2, $h\theta$ denotes the final hidden state of the MDM backbone, which is input to the unmasking head $f_\theta$ and the per-token quality head $g_\theta$. Why are they parameterized by the same parameters $\theta$? Could the authors provide a clear description on the whole structure of all the modules?

1. In Algorithm 1, how many forward passes are there in one iteration? In other words, is $f_{sg(\theta)}(\cdot|z)$ actually $sg(f_{\theta}(\cdot|z))$, which reuse $(f_{\theta}(\cdot|z)$ with the stop-gradient operation? Or is $f_{sg(\theta)}(\cdot|z)$ an independent forward pass that applies the stop-gradient operation first to the parameters $\theta$? According to the right subfigure of Figure 1, there seems to be two forward passes in one iteration? If so, why not just reuse $(f_{\theta}(\cdot|z)$?

1. In the right subfigure of Figure 1, where is the MDM loss?

1. According to Section 3.1, the quality score is computed only for a subset of clean tokens. Why not compute scores for all clean tokens? What if all the tokens from this subset have high scores?

1. What is the definition of "true per-token quality"?

1. By reading the current manuscript, I do not fully understand how the theoretical part has a strong connection and support to the ideas and claims in this paper. Could the authors provide detailed explanations on this point?

1. What is the most important difference between PRISM and the previous self-correction methods like [1,2]? The authors claim that the limitation of [1,2] is that they leverage the per-token unmasking posterior, not the joint posterior across positions. However, I do not see how PRISM overcomes this limitation from this paper.

1. This paper only compares PRISM with vanilla MDM and ReMDM, which are both training-free baselines. However, PRISM is a training-based method, and other training-based self-correction methods like P2-Train[1] are not included. How is PRISM compared with those training-based methods?

1. The authors claim that another limitation of methods like [1,2] is that they require an extra forward pass for scoring. As far as I know, the planner in [1] is a lightweight module and can be only 8M, which will not introduce too much overhead. Besides, PRISM introduces an additional module for per-token quality scoring and a LoRA adapter, which also brings overhead. Could the authors provide the overhead comparison between PRISM and methods like [1,2]?

### Reference
[1] Peng, Fred Zhangzhi, et al. "Path planning for masked diffusion model sampling." arXiv preprint arXiv:2502.03540 (2025).

[2] Gou, Zhibin, et al. "CRITIC: Large Language Models Can Self-Correct with Tool-Interactive Critiquing." The Twelfth International Conference on Learning Representations.

**Limitations:**

1. Comparisons with strong baselines are missing. This paper only compares PRISM with vanilla MDM and ReMDM, which are both training-free baselines. However, PRISM is a training-based method, and other training-based self-correction methods like P2-Train[1] are not included. The authors claim that there are different limitations in those previous methods, but the advantages of PRISM are not well supported by fair comparisons.

1. Some claims by the authors are not well supported. For example, the authors claim that another limitation of methods like [1,2] is that they require an extra forward pass for scoring. As far as I know, the planner in [1] is a lightweight module and can be only 8M, which will not introduce too much overhead. Besides, PRISM introduces an additional module for per-token quality scoring and also brings overhead.

1. The theoretical part does not show a strong connection and support to the ideas and claims in this paper.

### Reference
[1] Peng, Fred Zhangzhi, et al. "Path planning for masked diffusion model sampling." arXiv preprint arXiv:2502.03540 (2025).

[2] Gou, Zhibin, et al. "CRITIC: Large Language Models Can Self-Correct with Tool-Interactive Critiquing." The Twelfth International Conference on Learning Representations.

**Strengths And Weaknesses:**

### Strengths
1. Self-correction generation is a critical problem in diffusion LLMs. The study of self-correction is meaningful.

### Weaknesses
1. Some denotations are confusing. Please see Questions.
1. The motivation behind the proposed method is not clear enough, and the contribution is marginal compared with previous self-correction methods.
1. The empirical evaluations are not extensive enough. Comparisons with strong self-correction baselines are missing. Please see the Questions and Limitations.
1. Overall, the whole structure of this paper is unorganized and lacks logical coherence, with inconsistent context and low connections across ideas, claims, and theoretical analysis.

---

> ### Author Rebuttal · Authors · 2026-03-31
>
> We thank the reviewer for the questions. We respectfully believe that several of the concerns arise from a misunderstanding of our notation and from conflating PRISM with P2-Train. We also want to state that the manuscript is not AI-generated. If any points remain unclear, we are happy to clarify them in the discussion period.
>
> ### **Notations (Q1, Q2, Q3)**
> Our notation distinguishes between the model’s input/output and its internal hidden states. In Section 3.2, we state that $f_\theta$ and $g_\theta$ are the head outputs, generated by taking a masked sequence $w$ as input, passing it through the MDM backbone to produce hidden state $h_\theta$, and routing that state through the unmasking and token quality heads, respectively.
>
> In summary, $f_\theta$ and $g_\theta$ are not themselves the heads; they are the composition of the pretrained backbone with the unmasking head and token quality head respectively. Therefore, in Algorithm 1, $f_\theta$ refers to the MDM backbone (unmasking head)’s output.
>
> ### **P2-Train (Q9, Q10, Q11)**
> We agree that comparison to a training-based self-correction method is important. However, we emphasize that PRISM and P2-Train are not minor variants; they are fundamentally different. Discussion in our manuscript referred to the then-available arXiv v4, which had no public code and left the practical trained planner unclear. The later v5 update (March 5) clarifies this point. In particular, v5 states that the practical planner is instantiated as $Cat(z^j, B_\phi^j(z))$, i.e., as a scorer on the sampled clean proxy z, not on the current partially masked state x_t.
> Therefore, for the P2-Train loss, the minimizer is
> $p(x_0^j = z^j \mid z)= \mathbb{E}_{x_t \mid z}\left[p(x_0^j = z^j \mid z, x_t)\right]$;
> $x_t$ is effectively marginalized out.
>
> Thus, PRISM and practical P2-Train optimize different objects: PRISM learns token quality on the partially masked remasking state, whereas P2-Train scores a sampled clean proxy $z$. Our point is not that PRISM models the joint distribution (Q9); we do not make such a claim. Rather, two methods take different inputs and therefore cannot calibrate the same target.
>
> Regarding Q11, our point is architectural rather than absolute cost: PRISM obtains both outputs from one shared backbone pass, whereas P2-Train requires a separate scorer. For a fair comparison, we implemented P2-Train with the same architecture and parameter budget as our adapter, using a clean-sequence corrector from the paper description. Under the same 10k-step budget on OpenWebText, it shows near-zero correlation with oracle per-token quality (https://ibb.co/ZpG6VDpb - green curve) and lower generation quality (https://ibb.co/ymNnL1ws).
>
> We additionally implemented an extended version where the adapter has access to both $x_t$ and $z$. Then, the correlation improves from $\sim 0.4$ at early decoding steps to $\sim 0.9$ at later steps. This suggests that masked-state access and backbone fine-tuning can help, as also suggested by Proseco (Schiff et al., 2026; Feb 12),  but also reinforces our point: once the scorer uses masked-state information, it already departs from practical P2-Train.
>
> ### **Loss design (Q4, Q6, Q7)**
> We define true per-token quality in Section 3.1 as  $g_i^\star(y)=p(x_i = y_i \mid y \oplus m_i)$, i.e., the conditional likelihood of token $y_i$​ given the rest of the context.
>
> To clarify, there are two forward passes in each training step, as also illustrated in Figure 1. First, the model computes $f_\theta(z)$, whose logits are used to sample $y$ (line 5). The stop-gradient sign in Algorithm~1 indicates that the logits used to sample $y$ do not backpropagate through $f_\theta(z)$; these same logits are then reused in the regularization loss (line 8). Second, the model performs another forward pass with $y$ as input in order to compute the PRISM loss (line 7).
>
> Restricting the loss to a subset of clean tokens is essential, not arbitrary. Proposition 3.1 is proved exactly for the case where $y$ is obtained by unmasking selected positions from $z$, under which the minimizer reduces to the desired per-token quality. If BCE were applied to all clean positions, this guarantee would not follow in the same way.
>
> Finally, “all tokens in the subset having high scores” is not an issue during training, since the objective calibrates scores to ground-truth targets rather than ranking positions. Ranking matters only at inference-time remasking.
>
> ### **Figure 1 (Q5)**
> The MDM loss is omitted from Figure 1, where we intend to illustrate the main PRISM mechanism. Later, in Section 3.2, we explicitly state that we add the MDM training loss as a regularization term.
>
> ### **Logical coherence (Q8)**
> We respectfully believe that the theoretical and empirical parts are tightly connected in our manuscript; the intended logic is: definition of the target quantity (sec 2, 3.1) -> minimizer guarantee (sec 3.2) -> practical implementation (sec 3.3) -> empirical validation (sec 4).

---

> > ### Author Rebuttal · Reviewer_yKha · 2026-04-04
> >
> > Thank the authors for the detailed responses, which provide important information, especially the additional discussion about P2-Train. Could the author further explain this figure: https://ibb.co/ZpG6VDpb?
> >
> > Besides, the notations are still confusing to me. According to the authors' response,
> >  > In summary, $f_\theta$ and $g_\theta$ are not themselves the heads; they are the composition of the pretrained backbone with the unmasking head and token quality head respectively. Therefore, in Algorithm 1, $f_\theta$ refers to the MDM backbone (unmasking head)’s output.
> >
> > However, in this paper, $f_\theta$ is defined as the MDM in many places, e.g.,
> > - Page 4
> >   > In contrast, an MDM $f_\theta$ predicts from the masked position itself
> >
> >   > As we will show, PRISM does not possess these limitations: it is plug-and-play with any pretrained MDM $f_\theta$
> > - Page 5
> >   > Assume a pretrained MDM $f_\theta$ is given
> >
> >   > For a given sequence, we respectively denote the two head outputs as $f_\theta$ and $g_\theta$
> >
> > Similar problems can also be found in $g_\theta$.
> >
> > To summarize, what is $f_\theta$? What is $f_\theta (\cdot \vert \cdot)$? What is $f_\theta^j (x^j \vert z)$? And what about the $g_\theta$ family? Could the authors provide rigorous and coherent definitions?
> > The confusing notations with incoherent definitions are one of the important reasons why I suspected this paper was AI-generated.

---

> > > ### Author Response · Authors · 2026-04-06
> > >
> > > # Explanation on the P2-Train figure
> > >
> > > For P2-Train, following the description in the paper, we train an adapter consisting of a single-layer attention module followed by a projection layer, while keeping the MDM backbone frozen. The adapter is used at inference time as follows:
> > >
> > >  (1) The current state $x_t$ is passed through the MDM backbone to produce a one-step prediction $\hat{x}_0$.
> > >
> > > (2) This $\hat{x}_0$ is then fed into the adapter, which outputs a per-position score.
> > >
> > > For the plot, we then compute the correlation between the adapter-predicted per-token quality and the (empirical) oracle per-token quality. We collect the quality score for each clean token along inference-time trajectories, and compare it to the corresponding token likelihood $f_\theta^i(y^i \mid y \oplus m_i)\approx p(x^i = y^i \mid y \oplus m_i)$ queried from the base model. In other words, we measure how well-calibrated the adapter’s score is relative to the ground-truth per-token quality.
> > >
> > > Therefore, this plot shows that the PRISM fine-tuned model is indeed empirically well-calibrated relative to the ground-truth per-token quality (**validating our claim that our theory is empirically meaningful**), while P2-Train targets a different quantity, as we emphasize in our first-phase rebuttal.
> > >
> > > For the extended variant (orange curve), we modify the second stage by feeding $\hat{x}_0$ back into the MDM backbone. In this setup, both the MDM backbone and the adapter are *jointly* trained. This still requires two model passes: one to obtain $\hat{x}_0$​, and one for scoring. The results show that using only $\hat{x}_0$​ as input (P2-Train) is insufficient to learn reliable per-token quality, but as suggested in Proseco, this signal becomes more reliable when backbone fine-tuning is allowed.
> > >
> > > # Notation
> > >
> > > In Section 2, we state
> > > > This unmasking posterior is the core modeling component in MDM and is parameterized by a neural network $f_\theta$, which takes $z$ as an input and outputs a tensor of shape $|V| \times L$. Its $i$-th column, $f_\theta^i(\cdot | z) \in \Delta(V)$, models the unmasking posterior $f_\theta^i(v | z) \approx p(x^i = v | z)$.
> > >
> > > Accordingly, $f_\theta$ refers both to the MDM itself and to its output distribution, since the output of a pretrained MDM is precisely the unmasking posterior. This usage is standard in deep learning papers: one often uses $f_\theta$ both to denote the neural network and, in equations, the tensor output given by that network.
> > >
> > > Additionally, in PRISM, we attach an additional head for per-token quality prediction, and we denote this new output by $g_\theta$. We again recall that $g_\theta$ denotes the composition of the pretrained backbone with the attached token-quality head, the output of which is a length-$L$ tensor (as our target ground-truth per-token quality is a scalar per-position).
> > > Regarding the MDM parameterization, the (i)-th column of the output is a posterior distribution over the vocabulary (as in any token-distribution predictor). This is why we write $f_\theta^i(\cdot \mid z)$; specifically, $f_\theta^i(v \mid z)$ denotes the (v)-th entry of the (i)-th column. We also note that this posterior-style formulation is standard in recent MDM literature. For example, LLaDA [1] describes its model as:
> > >
> > > > the core of LLaDA is a mask predictor, a parametric model $p_\theta(\cdot \mid x_t)$ that takes $(x_t)$ as input and predicts all masked tokens ... and is trained using a cross-entropy loss ... $\log p_\theta(x_0^i \mid x_t)$.
> > >
> > > Likewise, in Train for the Worst, Plan for the Best [2], the denoising model is written in posterior form as $p_\theta(x_0^i \mid x_t, t)$. For this reason, we do not believe the manuscript’s original notation was inherently unclear.
> > >
> > > # Summary
> > > Since this reply will be the last publicly available response, and as this discussion was on the lengthier side, we would like to summarize by explicitly reiterating three broader points explicitly here, while noting that we are happy to address any remaining questions in discussion with the AC:
> > >
> > > **1**. This paper was entirely written by the authors, and our rebuttal has clarified that the manuscript’s notation is internally consistent.
> > >
> > > **2**. We continue to believe that the paper’s motivation, theory, and method are presented clearly overall, especially given that the reviewer did not raise any further questions about the paper’s core claims in the rebuttal acknowledgment.
> > >
> > > **3**. Regarding the paper’s empirical claim, we do not believe it is accurate to say that the evaluations are “not extensive enough,” especially because the suggested point of comparison was a fundamentally different method whose implementation details were not publicly available during the submission period.
> > >
> > > We thank the reviewer again for taking the time to engage with our work.
> > >
> > > [1] Large Language Diffusion Models, Nie et al, 2025.
> > >
> > > [2] Train for the worst, plan for the best: Understanding token ordering in masked diffusions, Kim et al, 2025.

---

### Official Review · Reviewer_XTsM · 2026-03-12

**Soundness:** 4
**Presentation:** 4
**Significance:** 3
**Originality:** 3
**Overall Recommendation:** 5
**Confidence:** 4

**Summary:**

This paper studies efficient token-quality estimation for self-correction in masked diffusion LLMs. Given a partially unmasked sequence $z$, the base model $f$ predicts masked-token distributions. The ideal quality score for an already unmasked token would be obtained by re-masking that token and evaluating  $f(\cdot|z\oplus m_i)$. Since doing this for every unmasked token is computationally expensive, the paper introduces an auxiliary head $g$ that predicts these per-token quality scores in a single forward pass. Overall, the contribution is a clean and practical approximation to token quality $f(x_i=z_i|z\oplus m_i)$ for remasking-based self-correction.

**Compliance With Llm Reviewing Policy:**

Affirmed.

**Final Justification:**

I appreciate the authors' thorough response. The additional clarifications and detailed explanations have resolved my initial concerns, leading me to raise my score.

**Key Questions For Authors:**

- Why is a separate head $g$ necessary? Could the base model $f$ be trained to directly predict the token probability at unmasked positions  $f(x_i=\cdot|z\oplus m_i)$? If it is computationally heavy, how hard it is to train such $f$?
- How reliable is the token-quality signal when the current sequence $z$ is still very low quality, such as in very early decoding steps?
- How sensitive are the results to the choice of $|S|$ or $|T|$? Are there ablations or theoretical guidelines for choosing them?
- Have authors considered adaptive remasking/unmasking policies based on global uncertainty or sequence-level confidence, instead of predetermined counts or distributions?

**Limitations:**

This paper did not provide limitation section. See key weaknesses.

**Strengths And Weaknesses:**

> Strengths

- The problem formulation is clean and easy to follow.
- The motivation is practical and well-grounded where exact token-quality computation would require many extra forward passes.
- The auxiliary head $g$ is a simple and efficient design choice.
- Proposition 3.1 is clear and helps formalize the target quantity.

> Weaknesses

- The remasking strategy may be too sensitive to local uncertainty. I kind of disagree that it cannot hurt. In examples like "I [mask] to school," a token such as "school" have low quality due to its many valid alternatives, but remasking it can still introduce unnecessary resampling cost even if changing it is not actually beneficial (since "school" already is a valid token).
- It is not yet clear why a separate scalar-quality head $g$ is preferable to training the base model $f$ itself to predict the probability $p(x_i=\cdot | z\oplus m_i)$ at unmasked position $i$. Author pointed out the scalar quality between 0 and 1 is simple and easy to learn, but I still think it is beneficial to see the entire token probability distribution. This is because it may help distinguishing between low quality token vs. valid token with many alternatives.
- The usefulness of token-quality estimates may degrade when the current sequence $z$ is globally poor (for example, when all positions are unmasked at first step), since the surrounding context is already very noisy. I understand this is orthogonal to the authors' main intent of approximating $g^i(z) \approx f(x_i=z_i|z\oplus m_i)$, but I think it is still an important question.
- Since usefulness of token quality depends on global quality, the choice of remasking and unmasking sizes, $|S|$ and $|T|$ seems crucial. I think fixed or sample from pre-specified distribution may be suboptimal relative to adaptive strategies.

---

> ### Author Rebuttal · Authors · 2026-03-31
>
> We thank the reviewer for the positive evaluation and for recognizing the novelty of our work, including our principled training objective (Proposition 3.1) and efficient training recipe. Below, we address the reviewer’s questions. If the questions are well-addressed, we kindly ask the reviewer to consider raising their score.
>
> ### **Can we instead learn the token probability?**
>
> The reviewer points out that, instead of learning our proposed per-token quality ( $f(x_i = z_i \mid z \oplus m_i) \in [0,1]$ ), one could fine-tune the model to learn the full per-position likelihood ( $f(x_i = \cdot \mid z \oplus m_i) \in \mathbb{R}^{|V|}$ ). One of the core intuitions behind PRISM is that a scalar per-token quality should already be sufficient for identifying low-quality or erroneous tokens (see lines 177–193), while being more training-efficient to learn than the full per-position likelihood, which is a distribution over the vocabulary. As suggested by the reviewer, below we discuss, both theoretically and empirically, the alternative of fine-tuning the model to learn the full per-position likelihood.
>
> *From a theoretical perspective*, our Proposition 3.1 extends naturally to learning the full likelihood: one can simply replace the binary cross-entropy with cross-entropy. In that case, the minimizer corresponds to the conditional distribution ( $f(x_i = \cdot \mid z \oplus m_i)$ ).
>
> *From a practical perspective*, we adapted the same OpenWebText setup used in our 170M-scale experiments and found that learning the $f(x_i = \cdot \mid z \oplus m_i)$ with the above CE loss shows lower generative quality than PRISM when both are trained for 10k steps (blue line).  (https://ibb.co/ymNnL1ws). However, the generative quality matches when trained for 20k steps (green line). This suggests that learning the likelihood may be more practical than we initially expected.  Moreover, learning the full per-position likelihood could enable more flexible remasking strategies, including Gibbs-style sampling; we agree that this is a promising direction.
>
> On the other hand, there is also an important scalability tradeoff. As model size grows, fully tuning an unmasking head might be inefficient to train. For larger-scale MDMs, such as LLaDA-8B, this efficiency advantage may still make PRISM’s scalar quality prediction preferable in practice.
>
> ### **Reliability of token quality**
>
> We would like to clarify that early decoding steps correspond to highly masked sequences but still in-distribution, rather than out-of-distribution. Since PRISM is explicitly trained on partially masked sequences across all masking ratios, there is no reason for its per-token quality estimates to become miscalibrated at early decoding stages.
>
> *Empirically,* we observe that the PRISM score remains well-calibrated with respect to oracle per-token quality throughout inference. In particular, as shown in Figure 6 in our manuscript, Pearson correlation remains above 0.9 across decoding steps, with especially strong correlation in early decoding, indicating that the signal remains reliable even when the sequence is largely masked.
>
> That said, the variance of the per-token quality can be higher in early decoding steps. As the reviewer pointed out, when most positions are still masked, low-quality predictions may reflect plausible alternatives rather than actual errors, making the estimates noisier, though still correlated with the oracle quality. In practice, this issue is less pronounced in conditional generation settings such as Sudoku or code generation, where the prompt already provides substantial context. Accordingly, in our experiments, we enable remasking only after sufficient context has been revealed (lines 796–801 for OpenWebText), and for LLaDA we allocate more remasking to intermediate decoding steps.
>
> ### **Choice of |S| and |T|**
>
> We provide ablations for [128,256,512] decoding steps with different choices of |T| on OWT, using our default setup where remasking is enabled only during the last half of decoding. We also evaluate a score threshold ($p_{th}$) based on a remasking strategy, which remasks clean tokens whose PRISM score falls below the threshold.
>
> Empirically, we find that overly aggressive remasking (large $\eta$ or $p_{th}$) often produces repetitive tokens. Under a fixed generation budget, more remasking induces a strong correction toward high-probability tokens, causing the model to collapse to common, predictable patterns. This typically results in lower sequence entropy and lower generative perplexity.
>
> While $p_{th}$ based remasking is a reasonable alternative for adaptively controlling the remasking rate during decoding, it did not outperform the simpler $\eta$ based strategy in our OWT experiments. We provide our results here: https://ibb.co/pBJ6nryw. Exploring more context-aware strategies, e.g., considering global uncertainty or sequence-level confidence, may be promising, but we consider this beyond the scope of our work.

---

> > ### Author Rebuttal · Reviewer_XTsM · 2026-04-02
> >
> > Thank you for the detailed clarification. The additional experiment on learning the full per-position likelihood is helpful. In fact, the result that it can match PRISM with longer training suggests that this alternative is more viable than the paper initially implied.
> >
> > My remaining concern is conceptual: for remasking decisions, I think token probability is important because it helps distinguish between genuinely wrong tokens and tokens that simply admit many plausible alternatives. A scalar quality score tends to conflate these two cases. This matters in examples like “I [m] to school”: even if remasking “school” may not hurt semantically, it can still incur unnecessary resampling cost and spend correction budget on a token that may already be acceptable.
> >
> > Similarly, the early-decoding discussion only partially addresses my concern. My issue is not whether partially masked inputs are in-distribution, nor whether PRISM remains correlated with the oracle target. Rather, I am not convinced we can tell whether a low-quality prediction in early decoding reflects an actual error or just uncertainty due to many plausible alternatives. In that regime, low quality alone may be an insufficient signal for remasking. This is exactly why I think the choice of $|S|$ and $|T|$ is central and should be handled more carefully, ideally with an adaptive policy based on more global or sequence-level signals.

---

> > > ### Author Response · Authors · 2026-04-06
> > >
> > > We thank the reviewer for the clarification. In this response, we provide a mechanistic analysis of the low-quality-but-high-entropy tokens and show that a likelihood fine-tuned model enables a more fine-grained remasking rule that addresses this issue. More broadly, we appreciate the reviewer for raising this perspective, while not viewing it as a fundamental limitation of PRISM, but rather as an interesting trade-off between a richer signal for improved remasking and training efficiency.
> > >
> > > Empirically, we identify the source of this phenomenon and show that a likelihood-based extension addresses it effectively in practice, and theoretically, this extension is also naturally supported by the PRISM guarantee.
> > >
> > > **PRISM guarantee extends to likelihood learning**: We first recall the point from our earlier rebuttal regarding the extension of the PRISM guarantee. Specifically, replacing the BCE objective with a CE objective yields a likelihood-learned model, whose minimizer corresponds to the conditional likelihood. Thus, the same theoretical guarantee naturally extends from per-token quality learning to per-token likelihood learning.
> > >
> > > **Mechanistic analysis**: Using this likelihood fine-tuned model, we compute the per-position entropy $H^i(y) \coloneqq -\sum_{v \in \mathcal{V}}p(x^i=v \mid y \oplus m_i) \log p(x^i=v \mid y \oplus m_i)$. The resulting plot (https://ibb.co/NzQFyfD ) confirms the reviewer’s intuition: low per-token quality can indeed arise at positions with many plausible alternatives, particularly in early decoding steps. We note again that our current decoding procedure already partially mitigates this issue by performing less remasking during early decoding steps.
> > >
> > > **Entropy-aware remasking**: At the same time, such a CE-loss fine-tuned model allows a more refined remasking rule that uses token entropy as an additional signal. Concretely, we consider a likelihood-aware scoring scheme based on the per-token distribution: $s^i(y) = p(x^i = y^i \mid y \oplus m_i) * \exp(H^i(y))$, which can be viewed as an entropy-compensated score that avoids over-penalizing high-entropy positions. For example, this prevents unnecessary remasking in cases such as “I [mask] to school”.
> > > |          |  |  | MAUVE |  |  |  |  | Gen PPL. |  |  |  |  | Entropy |  |  |
> > > |----------------|-------|-------|-------|-------|-------|----------|----------|----------|----------|----------|---------|---------|---------|---------|---------|
> > > | NFE            | 64    | 128   | 256   | 512   | 1024  | 64       | 128      | 256      | 512      | 1024     | 64      | 128     | 256     | 512     | 1024    |
> > > | Baseline   | 0.144 | 0.249 | 0.359 | 0.435 | 0.421 | 18.6     | 15.2     | 13.7     | 13.0     | 12.7     | 5.20    | 5.15    | 5.11    | 5.09    | 5.08    |
> > > | $s^i(y)$     | **0.195** | **0.322** | **0.433** | **0.442** | **0.520** | 20.7 | 18.4 | 17.5 | 17.1 | 16.9 | 5.28 | 5.25 | 5.22 | 5.21 | 5.20 |
> > >
> > > Under this new score, we evaluate generation quality under the same hyperparameter settings and the same $\eta$-based remasking strategy. With the same underlying model, incorporating per-token entropy consistently improves generation quality (measured by MAUVE) across decoding steps compared to the baseline that relies only on scalar per-token quality. In addition, the corresponding entropy-versus-quality scatter plot shows that remasking becomes naturally concentrated on **low-entropy, low-quality** tokens. (https://ibb.co/RGfY4PgB )
> > >
> > > We therefore agree with the reviewer that a scalar per-token quality head alone is insufficient to capture this distinction, as our new mechanistic analysis and refined remasking algorithm suggest. Moreover, our results show that per-token likelihood is already sufficient to effectively filter high-entropy, low-quality tokens, **without requiring** additional global or sequence-level signals.
> > >
> > > That being said, this richer supervision comes with a trade-off: on OWT, we observe roughly a **2× training efficiency gap** between learning per-token quality and learning per-token likelihood; it remains unclear how this gap would evolve at larger scales, where likelihood learning may become less efficient.
> > >
> > > Overall, we thank the reviewer for highlighting this point. We instead view this as an interesting trade-off between richer signal (for better remasking algorithms) and training efficiency, rather than a fundamental limitation of the current formulation: **empirically**, we identify the source of this phenomenon and show that a likelihood-based extension can effectively address it; **theoretically**, this extension is naturally supported by the PRISM guarantee. We would be happy to clarify this discussion in the revised manuscript.
> > >
> > > **If the questions are well-addressed, we kindly ask the reviewer to consider raising their score.**

---

### Official Review · Reviewer_ZVrA · 2026-03-12

**Soundness:** 4
**Presentation:** 4
**Significance:** 3
**Originality:** 4
**Overall Recommendation:** 5
**Confidence:** 4

**Summary:**

The authors propose PRISM as a method to self-correct masked diffusion models (MDMs) for language. PRISM works by training a lightweight head onto a pretrained MDM backbone to predict which positions should be remasked. They show that unlike other self-correction techniques, PRISM is well defined via its loss function and does not depend on the quality of the pretraind MDM denoiser. They benchmark PRISM across multiple MDM models and benchmarks with strong experiments.

**Compliance With Llm Reviewing Policy:**

Affirmed.

**Final Justification:**

This paper proposes PRISM, a method for self-correction in masked diffusion models that trains a lightweight head on a pretrained MDM backbone to predict which positions should be remasked. The paper's core strength is its principled theoretical foundation.

**Soundness.** The theoretical contributions are excellent: defining a proper notion of self-correction via a target quantity (per-token quality), deriving a loss function whose minimizer achieves this target, and crucially showing that the loss does not depend on the quality of the pretrained denoiser. This independence from the denoiser is a meaningful property that distinguishes PRISM from heuristic approaches. The limitation to local corrections is acknowledged and remains an open problem for the field.

**Originality.** The formulation is novel — grounding self-correction in a well-defined loss function rather than relying on heuristic confidence scores is a clear conceptual advance. The algorithmic design (data-efficient training, cheap inference) makes the method practical.

**Significance.** Self-correction is an important capability for MDMs, which are prone to error accumulation due to parallel token generation with independent position-wise predictions. PRISM provides a principled and practical solution with strong experimental results across multiple models and benchmarks.

**Clarity.** The paper is well-written with clear exposition of the theoretical framework and experimental setup.

The rebuttal addressed all my concerns. The authors provided a helpful clarification that the accuracy gains of ReMDM baselines in Figure 2 stem from continued cross-entropy training (used as a regularizer), not from improved correction capability — an important distinction I had not fully appreciated. They also agreed to amend the sample-efficiency claim and to include discussion of contemporary self-correction works. I maintain my score of 5 (Accept).

**Key Questions For Authors:**

1. Can you please add a section for contemporary related works and compare with the following recent papers? Feel free to do so in the main body of the paper or the appendix. They seem quite relevant to discussion of self-correction.
- Schiff, Yair, et al. "Learn from Your Mistakes: Self-Correcting Masked Diffusion Models." arXiv preprint arXiv:2602.11590 (2026).
- Bie, Tiwei, et al. "LLaDA2. 1: Speeding Up Text Diffusion via Token Editing." arXiv preprint arXiv:2602.08676 (2026).

2. Please fix the minor typos:
- On line 131 you write `replace it with a new clean` instead of `replace it with a new clean token`
- On 406 you write $x_t s$, which I believe has an extra $s$ term

3. On line 365-369 you claim: "PRISM starts to outperform vanilla MDM inference and baseline within only a few epochs, far fewer than pertaining epochs, corroborating our claim of sample efficiency". Although true, I think this is not a surprising statement since all self-correction methods that build upon a pretrained denoiser are quite likely sample-efficient. In fact, Figure 2 shows that RemDM-conf and RemDM-cap share this behavior. As such, can you please amend this claim to acknowledge that this sample efficiency is shared by other similar methods and is not unique to PRISM?

**Limitations:**

Yes

**Strengths And Weaknesses:**

Strengths:
1. Very strong theoretically relevant insights:
- Define what a proper notion of self correction should look like i.e. $g_*^i (y) = p(x^i = y^i | y \oplus m_i)$
- Define a loss function that achieves $g_*^i$ as its minimizer and does not depend on the pretrained denoiser $f_theta^I$
2. Strong experiments that show good performance across a variety of models and benchmarks
3. Strong algorithmic components: data efficient to train, cheap to use during inference

Weakness:
1. PRISM is only capable of making local self corrections, not global. This might lead to degenerate behavior such as repeatedly unmasking then masking a position to the same token value.

---

> ### Author Rebuttal · Authors · 2026-03-31
>
> We thank the reviewer for the positive evaluation and for recognizing the novelty of our work, particularly the principled formulation of the target quantity (per-token quality), our method for learning it, and the experimental results. Below, we address the reviewer’s comments and questions.
>
> ### **Discussion of related work**
>
>  We note that the two related works mentioned by the reviewer became publicly available only after the submission deadline. They are indeed closely related to our framework in that they also train a model, beyond an initial MDM objective, to predict token quality for clean tokens. We would be happy to include a thorough methodological comparison in the revised version of the manuscript.
>
>
> ### **Concern on the global correction**
>
>  The reviewer is right that our current framework primarily handles local corrections. To clarify, enabling global correction mechanisms remain an outstanding open question in this field, namely discrete diffusion models, as we note in the conclusion of our manuscript.
>
> ### **Typos**
>
>  We thank the reviewer for pointing out the typos and will make sure to correct them in the revised version.
>
> ### **Clarification of the sample-efficiency claim**
>
> Our main point regarding sample efficiency is the following: unlike standard fine-tuning, where the goal is to adapt a large pretrained model to new tasks while preserving its original capabilities, our setting asks the model to learn a new notion of per-token quality that it has not encountered during pretraining. In that sense, it may not have been obvious that this new capability could also be learned sample-efficiently. That said, as the reviewer points out, this efficiency is also shared by training-free self-correction methods such as ReMDM. We will revise the manuscript to make this claim more precise.
>
> We clarify that the accuracy gains of ReMDM-cap and ReMDM-conf in Figure 2 do not reflect improved error correction capability. Instead, they primarily result from continued training under the vanilla MDM’s cross-entropy objective, which we use as a regularizer. We include these results for fair comparison, emphasizing that the observed improvements of PRISM arise from its new correction ability rather than enhanced demasking ability.

---

> > ### Author Rebuttal · Reviewer_ZVrA · 2026-04-01
> >
> > We thank the authors for their response that address all our concerns.
> >
> > We acknowledge that the papers we requested to be cited were released after the submission deadline, and thus are not mandatory to include and compare against. However, if the authors are willing, we think it would help clarify PRISM in the context of additional contemporary papers. We ultimately leave this to the discretion of the authors.

---

> > > ### Author Response · Authors · 2026-04-07
> > >
> > > We thank the reviewer again for the positive evaluation. We are happy to include the comparison to the two papers (Proceso, LLaDA 2.1) in our further version of the manuscript.

---

### Official Review · Reviewer_JFiL · 2026-03-14

**Soundness:** 3
**Presentation:** 3
**Significance:** 4
**Originality:** 2
**Overall Recommendation:** 4
**Confidence:** 3

**Summary:**

The paper introduces a fine-tuning framework designed to equip pretrained Masked Diffusion Models (MDMs) with self-correction capabilities. While the vanilla MDMs lock in tokens once they are unmasked, preventing them from fixing early dependency errors. Existing works try to fix this issue, with simple heuristics, while this work improves upon them by fine-tuning a simple adapter that predicts the likelihood of a token given the surrounding context allowing the model to dynamically detect and remask low-quality tokens during inference. The authors demonstrate empirical gains across Sudoku, unconditional text generation (170M), and code generation (LLaDA 8B).

**Compliance With Llm Reviewing Policy:**

Affirmed.

**Final Justification:**

The rebuttal adresses my minor concerns, while the major one regarding the computational complexity remains. Therefore I retain my original recommendation still arguing that the proposed work quite cleverly tackles a fundamental limitation of current masked diffusion models

**Key Questions For Authors:**

None

**Limitations:**

Yes, limitations are properly discussed

**Strengths And Weaknesses:**

Strengths
- This paper tackles a fundamental limitation of current Masked Diffusion Models (MDMs) - their inability to self-correct or remask tokens once they are generated. Solving this "lock-in" problem is a highly important and promising direction for improving the reliability and reasoning capabilities of discrete diffusion models.
- A major strength of PRISM is its practical application. Rather than requiring a complete architectural overhaul or expensive pretraining from scratch, the method ingeniously attaches a trainable adapter to an already pre-trained MDM. This makes the approach easy for the broader community to adopt on top of existing foundation models, although I’m not particularly convinced about this idea being “lightweight” - see weaknesses
- The idea for calculating the loss through the marginalization perspective is clever and avoids the flaws of other approaches (e.g distillation)
I find it interesting how this idea can be related to work by Nichol & Dhariwal (2021) [1] . In that paper, a similar approach was used where a secondary output predicted the reverse process variance instead of relying on fixed constants. Here, the authors successfully adapt this concept to masked diffusion by adding a trainable adapter to predict per-token quality.

[1] Nichol, Alexander Quinn, and Prafulla Dhariwal. "Improved denoising diffusion probabilistic models." International conference on machine learning. PMLR, 2021.


Weaknesses
- The authors frame PRISM as a "lightweight" plug-in, but fine-tuning the 250M parameter adapter for the 8B LLaDA model still required ~360 H100 GPU hours and a curated dataset of 100k pairs. Compared to purely algorithmic, training-free methods like ReMDM (which require zero GPU hours or extra data), this computational and data barrier makes PRISM significantly less "plug-and-play."
- On benchmarks like HumanEval, standard AR models require only as many steps as the generated output length (typically 50–200 steps) and leverage KV-caching to make each step computationally cheap. In contrast, PRISM requires up to 1024 sampling steps to achieve its peak 42.7% accuracy on HumanEval. Because masked diffusion requires a full, dense forward pass over the entire sequence at every step, PRISM's inference compute and latency are astronomically higher than a standard AR model, limiting its real-world deployability.

Small issues:
- Please correct me if I’m wrong, but while the adapter is a relatively small network comparing to the large backbone model, using it is not completely free, so it’s not really precise that the “method does not add any computational overhead” - I agree that it’s negligible, but this just brings confusion

---

> ### Author Rebuttal · Authors · 2026-03-31
>
> We thank the reviewer for the positive evaluation and for recognizing the novelty of our work, especially in its practical applicability and loss design. Below, we provide additional comments on the reviewer’s points.
>
> ### **Connection to Nichol & Dhariwal (2021) [1]**
>
> We appreciate the reviewer for pointing us to this paper. Our work is similarly motivated by the idea that leveraging a “secondary output” in a clever algorithmic way can lead to improved generative inference.
>
>
> ### **Comparison to autoregressive models**
>
> We note that systems-level efficiency advantages of autoregressive LLMs, such as KV caching and highly optimized left-to-right decoding, are largely orthogonal to the main focus of this work. Our goal here is not to resolve the inference-efficiency gap between diffusion and autoregressive models, but to study whether diffusion LLMs can benefit from a principled self-correction mechanism. In that regard, PRISM is particularly well aligned with diffusion models, since it enables revision directly on a fixed canvas via targeted remasking and regeneration of specific positions, rather than only through continuing generation from a fixed prefix.
>
>
> ### **Clarification of the “no computational overhead” claim**
>
>  To clarify, our method adds LoRA adapters to the transformer backbone together with an auxiliary remasking head. By “no computational overhead,” we meant that for a given partially masked sequence, a single forward pass through the backbone (equipped with the LoRA adapters) produces both (1) predictions for masked positions and (2) per-token quality estimates for clean positions. Of course, this does introduce additional adapter parameters, so our claim was not about parameter count and training cost, but rather about inference-time computation per forward pass. We will revise the manuscript to make this distinction explicit and to clarify that “no computational overhead” refers to obtaining both quantities in a single forward pass.

---

> > ### Author Rebuttal · Reviewer_JFiL · 2026-04-01
> >
> > Thank you for the response, regarding the no computational overhead, I was just referring to the precise wording. I got confused whether I understand the idea properly if it brings "no computational overhead".
> >
> > The main weaknesses I raised in the review are of the more general manner and cannot be resolved in a short rebuttal part. Therefore I decide to keep my original score of weak accept

---

> > > ### Author Response · Authors · 2026-04-07
> > >
> > > We thank the reviewer again for the positive evaluation. In this response, we provide additional perspective to clarify how to interpret this limitation and position our contribution within the broader landscape. As noted by the reviewer and as we acknowledge in the conclusion, PRISM corrects errors via per-position likelihood, and thus does not explicitly capture global, sequence-level errors. Enabling global self-correction remains an important open challenge in this field.
> > >
> > > That being said, as the reviewer also points out, our framework introduces a complementary perspective based on marginalization, which enables a new axis of fine-tuning through per-position signals without requiring an explicit verifier. In contrast, existing approaches to global self-correction in LLMs typically rely on reinforcement learning with sequence-level rewards, which provide relatively coarse supervision and often require full generation trajectories, e.g., [1].
> > >
> > > From this viewpoint, PRISM offers a more fine-grained and local learning signal that is largely orthogonal to RL-based approaches. We therefore see these directions as complementary: some errors are inherently local and can be addressed through per-token calibration, while others are global and may benefit from sequence-level optimization. Exploring how to combine these two axes for more effective self-correction is an exciting direction for future work.
> > >
> > > [1] Training Language Models to Self-Correct via Reinforcement Learning, Kumar et al, 2024.

---

### Decision · Program_Chairs · 2026-04-30

**Decision:**

Accept (regular)

**Comment:**

There is unanimous agreement that self-correction in MDLM is an important problem and that the paper makes an original contribution based on fine-tuning a pre-trained MDLM. Also as one reviewer noted that " idea for calculating the loss through the marginalization perspective is clever and avoids the flaws of other approaches", this is clearly an important contribution.

 The major concern of the reviewer who gave a Reject was lack of comparison with P2-Train, to which the authors adequately responded.